# Cocaine self-administration induces sex-dependent protein expression in the nucleus accumbens

Alberto J. López[1,2,3,13], Amy R. Johnson[1,2,3,13], Tanner J. Euston[4,5], Rashaun Wilson[6,7], Suzanne O. Nolan[1,2,3], Lillian J. Brady [1,2,3], Kimberly C. Thibeault [1,2,3], Shannon J. Kelly[1,2,3], Veronika Kondev[1,2,3], Patrick Melugin[1,2,3], M. Gunes Kutlu[1,2,3], Emily Chuang[1,2,3], TuKiet T. Lam [6,7,8], Drew D. Kiraly[4,5,9,10] & Erin S. Calipari [1,2,3,11,12✉]

Substance use disorder (SUD) is a chronic neuropsychiatric condition characterized by long-lasting alterations in the neural circuitry regulating reward and motivation. Substantial work has focused on characterizing the molecular substrates that underlie these persistent changes in neural function and behavior. However, this work has overwhelmingly focused on male subjects, despite mounting clinical and preclinical evidence that females demonstrate dissimilar progression to SUD and responsivity to stimulant drugs of abuse, such as cocaine. Here, we show that sex is a critical biological variable that defines drug-induced plasticity in the nucleus accumbens (NAc). Using quantitative mass spectrometry, we assessed the protein expression patterns induced by cocaine self-administration and demonstrated unique molecular profiles between males and females. We show that 1. Cocaine self-administration induces non-overlapping protein expression patterns in significantly regulated proteins in males and females and 2. Critically, cocaine-induced protein regulation differentially interacts with sex to eliminate basal sexual dimorphisms in the proteome. Finally, eliminating these baseline differences in the proteome is concomitant with the elimination of sex differences in behavior for non-drug rewards. Together, these data suggest that cocaine administration is capable of rewriting basal proteomic function and reward-associated behaviors.

[1] Department of Pharmacology, Vanderbilt University, Nashville, TN, USA. [2] Vanderbilt Center for Addiction Research, Vanderbilt University, Nashville, TN, USA. [3] Vanderbilt Brain Institute, Vanderbilt University, Nashville, TN, USA. [4] Department of Psychiatry, Icahn School of Medicine at Mount Sinai, New York, NY, USA. [5] Friedman Brain Institute, Icahn School of Medicine at Mount Sinai, New York, NY, USA. [6] Department of Molecular Biophysics and Biochemistry, Yale University, New Haven, CT, USA. [7] WM Keck Biotechnology Resource Laboratory, Yale University, New Haven, CT, USA. [8] Yale/NIDA Neuroproteomics Center, New Haven, CT, USA. [9] Nash Family Department of Neuroscience, Icahn School of Medicine at Mount Sinai, New York, NY, USA. [10] Seaver Center for Autism, Icahn School of Medicine at Mount Sinai, New York, NY, USA. [11] Department of Molecular Physiology and Biophysics, Vanderbilt University, Nashville, TN, USA. [12] Department of Psychiatry and Behavioral Sciences, Vanderbilt University Medical Center, Nashville, TN, USA. [13]These authors contributed equally: Alberto J. López, Amy R. Johnson. ✉email: erin.calipari@vanderbilt.edu

Substance use disorder (SUD) is a debilitating neuropsychiatric disorder characterized by repeated drug use which induces plasticity within the brain to drive continued drug taking and seeking. Although both males and females can develop SUD, sex differences in drug taking and SUD vulnerability for stimulant drugs of abuse exist in both human and animal models[1–5]. In the human literature, although males are more likely to be given the opportunity to try cocaine than females, both sexes progressed to regular at an equal rate once exposed. Further, when opportunity for drug use is taken into consideration, female use of psychostimulants exceeded that of males and females transitioned from recreational use to SUD at a faster rate[6–8]. Similar sex differences are observed in animal models, suggesting key biological differences between males and females likely underlie these effects[9–11]. Indeed, in addition to behavioral differences, males and females also show different pharmacological responses to various drugs of abuse including cocaine, nicotine, and opioids, which may lead to different drug-induced neural plasticity - either in the mechanism or relative magnitude of these effects[5,8].

While a large amount of work has focused on sex differences in the behavioral and physiological effects of drugs of abuse, we also still lack an understanding of the wide-spread molecular differences in the brain that exist between males and females at baseline. It is likely that differences in protein expression at baseline interact with drug actions on reward systems to alter drug-induced molecular plasticity and may provide a framework that can explain sex differences in both vulnerability and the trajectory of SUD. Understanding the interaction between these factors is particularly relevant as significant work has suggested that males and females have different behavioral strategies, especially as it relates to reward and reinforcement behavior[12]. These sex-specific behavioral strategies have been largely linked to reward-related brain regions like the nucleus accumbens (NAc), where manipulations of this area (either directly or via manipulations of projections into the NAc) are capable of blocking/inducing these differences in behavior[12–14]. Thus, a major goal of this study was to focus on basal sex differences in motivation and protein expression in the NAc, and how these interact with cocaine self-administration to dictate the molecular and behavioral adaptations that occur following cocaine exposure.

A particular focus of research on drug-induced protein regulation has been the adaptations identified in the NAc, given its critical role in motivation and reward seeking. However, when studying adaptations within the NAc, the field has focused largely on single protein targets or large-scale transcriptional adaptations in response to drug-induced plasticity—typically only in male animals[15–18]. While this mechanistic single-target approach has been incredibly effective, data-driven approaches pose an advantage in that the complexity seen between the sexes may be represented in large-scale expression patterns rather than at the single gene/protein level. Moreover, large-scale "omic" studies allow for the identification of expression patterns—especially as they relate to interactions between basal differences and stimulus-evoked changes—providing a comprehensive framework for understanding how these differences emerge. This is particularly important as it relates to understanding sex differences as it is likely that basal proteomic differences may dictate sex-specific cocaine responses. Quantitative proteomics—which we employ here—provides an advantage over typically used transcriptomics in that changes in protein expression levels are more directly linked to alterations in cellular function. It is also possible that similar behaviors (cocaine reinforcement) may be driven largely by different molecular effectors in males and females, an effect that would be critical to understand to develop efficacious pharmacotherapies for both sexes[19].

Here, we combine large-scale proteomic analysis of the NAc with complex reinforcement behavior for drug and nondrug reinforcers to characterize the molecular mechanisms with the potential to regulate sex-specific reward behaviors in male and female mice. Importantly, while we find some overlap in cocaine-induced protein expression with threshold-free approaches, we find almost nonoverlapping molecular plasticity in the proteins significantly regulated by a history of cocaine self-administration between males and females. Interestingly, the effects of cocaine seemed to be greater on proteins that were sexually dimorphic at baseline, where it acts to reduce the sexual dimorphisms that were observed both in the proteome and in motivated behavior.

## Results

**Females are more motivated to self-administer sucrose.** To understand whether there were differences in reinforcement between males and females at baseline, we first conducted a series of studies to test reward consumption and motivation under various conditions (Fig. 1a). When mice were sated through ad libitum access to food in the home cage, females consumed more sucrose than males (as a function of their body weight) under a fixed ratio 1 (FR1) schedule of reinforcement (10 μL/sucrose delivery; Fig. 1b; unpaired two-tailed $t$ test: $t_{16} = 2.63$, $p = 0.018$). However, when schedules were changed from variable ratio 3 (VR3) to variable ratio 5 (VR5) so that the effort required to self-administer sucrose was higher, female mice increased sucrose consumption under these conditions, while male mice trended toward decreased consumption (plotted as a function of body weight; unpaired two-tailed $t$ test: $t_8 = 3.26$, $p = 0.012$; Fig. 1c, d). This suggested that males and females may be differentially sensitive to changes in cost. To directly test this, we used a behavioral economics approach in rats where consumption is determined as a function of price (in this case, price = number of responses required to obtain one sucrose pellet), using the within-session threshold procedure (Fig. 1e). We found that the maximal price that rats are willing to pay ($P_{max}$) for sucrose is higher in female rats as compared to male rats (Fig. 1f–h), while how much sucrose they consumed at a minimally constraining price did not differ (Supplementary Fig. 1). Together, these data show that there are sex differences in motivation for nondrug rewards with females exhibiting higher motivation than males. Even more importantly, these data indicate that these differences in motivation and sensitivity to relative cost are conserved across rodent species.

**There are robust sex differences in proteins linked to reward, motivation, and cocaine responses in the NAc.** Changes in reinforcement and motivation have been linked to reward-related brain regions, specifically the NAc. In fact, previous work from our group has shown sex differences in the neural circuitry of the NAc that has been linked to reward-learning and motivation, making this region a prime target for the identification of molecular targets that underlie these sex differences[12,20,21]. However, while differences in the expression of single genes and proteins have been identified between males and females, it is important to understand how patterns that emerge across the entire proteome contribute to these effects.

To this end, we harvested NAc tissue from male and female mice and, using data-independent acquisition (DIA) mass spectrometry, measured protein expression between males and females. Of 1496 robustly detected proteins (Supplementary Data 1), 112 proteins were differentially expressed in a sex-specific fashion (Fig. 2a and Supplementary Data 2). Within these sexually dimorphic proteins, we identified proteins previously implicated in reward- and drug-associated responses, such as

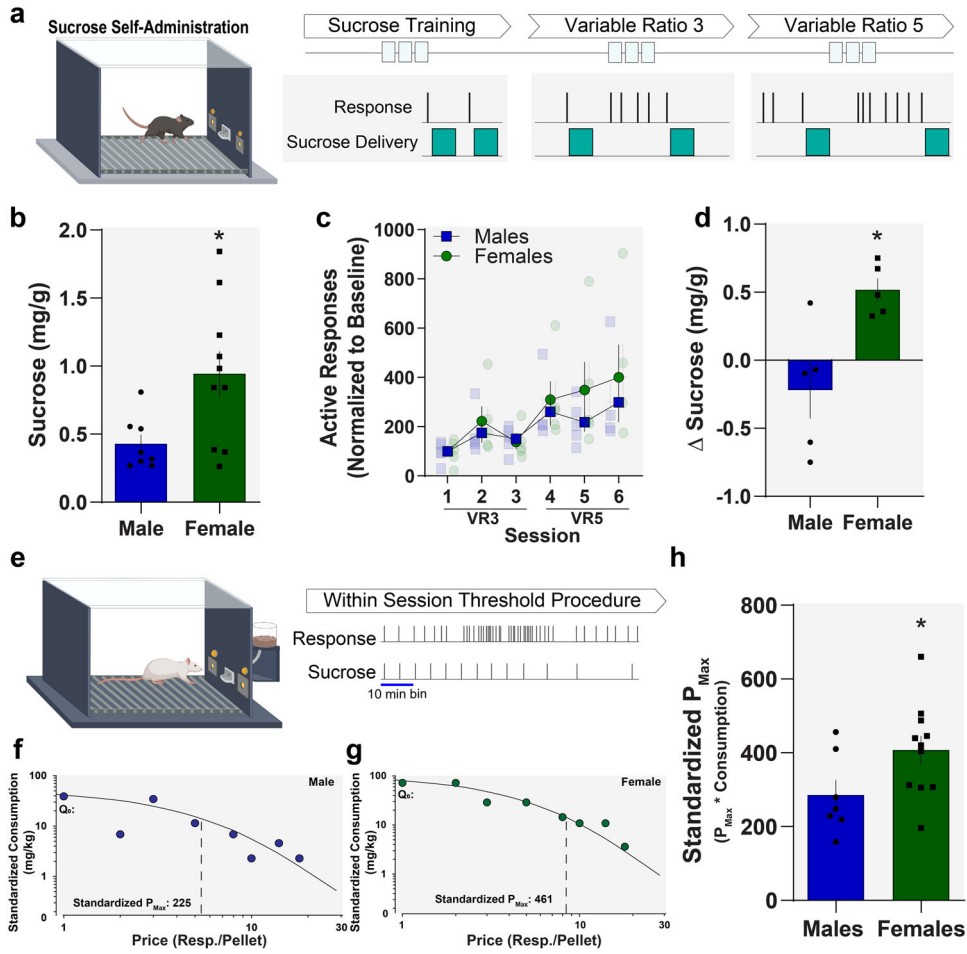

**Fig. 1 Sexual dimorphisms in motivation: female mice work harder for nondrug rewards. a** Schematic of sucrose self-administration in male and female mice. Mice went through a series of different reinforcement schedules to determine how their behavior changed with changing contingencies. **b** Average sucrose consumption under a fixed ratio 1 (FR1) schedule of reinforcement (1 response = 1 sucrose delivery) between male and female mice in a sated condition, normalized to body weight. Females consume more sucrose than males. **c** Average responses for sucrose under variable ratio 3 (VR3) and VR5 schedules of reinforcement between male and female mice, normalized to baseline (responses during first day of VR3). **d** Average change in sucrose consumption in mg from VR3 to VR5 between male and female mice, normalized to body weight. Female mice increase consumption compared to male mice when the required effort to obtain each reinforcer was increased. **e** Schematic of within-session threshold procedure in male and female rats. Representative demand curves for sucrose of **f** male and **g** female rats during the within-session threshold procedure. **h** Economic analysis: standardized $P_{max}$ values—the maximal price an animal is willing to pay in effort to obtain sucrose—for male and female rats during the within-session threshold procedure. Standardized $P_{max}$ is calculated as $P_{max} \times Q_0$ and controls for potential differences in consumption between groups. We did not find differences in sucrose consumption between males and females under these conditions (Supplementary Fig. 1). Standardized $P_{max}$ for sucrose is higher in females compared to males indicating that they are more motivated to work to obtain sucrose as effort requirements are increased. All data reported as mean ± SEM. *$p < 0.05$. Diagrams created with biorender.com.

GRM2[22], VPS51[23], SATT[24], VGLU3[25], IPO4[26], VAMP1[27], and CYFP2[28] (Fig. 2b). We also show that some of these targets are also elevated at the transcriptional level (Supplementary Fig. 2). To characterize the cellular functions of differentially expressed proteins, we used Gene Ontology (GO) analysis, and identified GO terms downregulated (Fig. 2c) and upregulated (Fig. 2d) in the NAc of female mice relative to male mice. Of note are the differentially enriched GO terms linked to neuronal signaling, including cytoplasmic part (Fig. 2e, top) and protein binding (Fig. 2e, bottom). Lastly, we characterize protein–protein functional networks using STRING to descriptively assess potential interactions in the proteins that were differentially regulated between males and females at baseline (Fig. 2f). Of particular interest to us based on our previous work were the functional networks involved in gene regulatory processes (such as the family of histone H2 proteins; Fig. 2g) and synaptic structure/plasticity (such as VAMP2/CYFP2/ACTB network;

Fig. 2h). These processes have been demonstrated to have critical roles in cocaine-associated behaviors and responses. As such, we hypothesized that previously reported differences between males and females in cocaine-associated behaviors are underlined by baseline differences in NAc proteomes and subsequent alterations in functional protein interactions.

**Sex differences in the motivation to self-administer cocaine.** Given that the protein targets that were identified at baseline have been associated with neural and behavioral responses to cocaine, we next wanted to understand how cocaine reinforcement differed between male and female mice. Previous work in rats and primates has demonstrated that there are sex differences in cocaine reinforcement, with females showing enhanced motivation for cocaine; however, these only emerge in specific contexts and studies have not comprehensively assessed these effects in mice[5,9,10,20,29]. To this end, we tested whether differences are also apparent in C57BL/6J mice

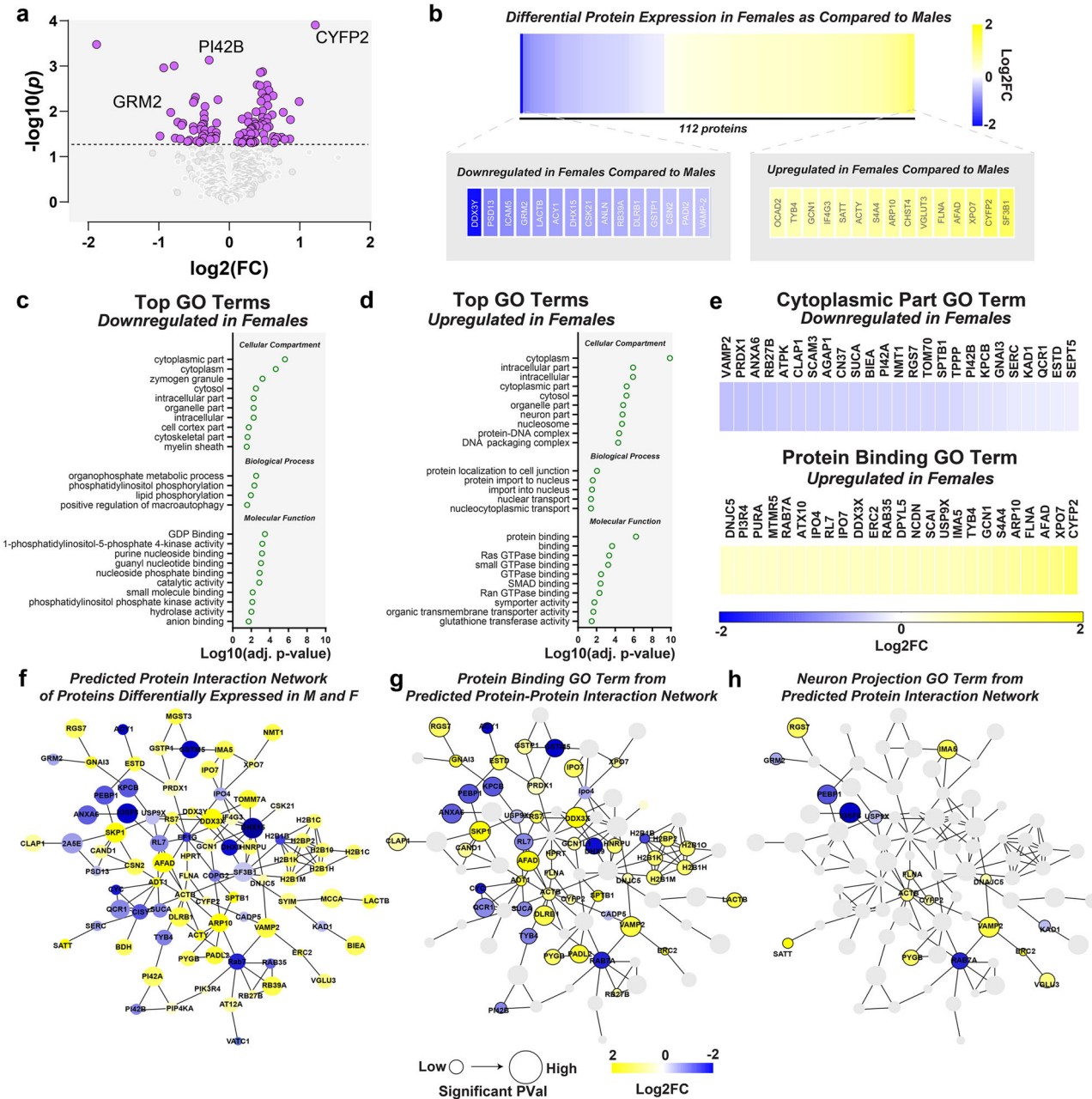

**Fig. 2 Sexual dimorphisms in the proteomic landscape of the nucleus accumbens.** Mass spectrometry was run on tissue from the nucleus accumbens (NAc) of male and female mice to determine the proteomic landscape. Differential expression between females as compared to males was assessed in control mice. **a** Volcano plot showing sexually dimorphic proteins in the nucleus accumbens from female versus male mice. Male and female accumbens proteomes diverge at key proteins—some associated with reward and drug response are noted. **b** Heat map of baseline sexually dimorphic proteins in the accumbens. Insets showing the top 15 proteins that were decreased (left) and increased (right) in females as compared to males. **c** In order to conduct Gene Ontology (GO) analysis protein names had to be converted to associated gene names. These conversions are available in Supplementary Data 1. Top GO terms in proteins downregulated in females (compared to males) and **d** proteins upregulated in females (compared to males). **e** Proteins making up the representative GO terms (top) downregulated in females (cytoplasmic part) and (bottom) upregulated in females (protein binding). **f** STRING analysis of predicted protein interaction network of proteins differentially expressed in males and females (**g**) within the protein binding cluster, and (**h**) neuron projection.

across a variety of paradigms (Fig. 3a and Supplementary Fig. 3a). Male and female mice acquired cocaine self-administration (at 1 mg/ kg/injection i.v.) at equal rates and consume similar levels of cocaine on low effort schedules (FR1), as well as higher effort fixed ratio schedules (FR3 and FR5; Fig. 3b; two-way repeated measures ANOVA: female: significant effect of lever [$F_{(1, 36)} = 92.60$, $p < 0.0001$; significant effect of schedule [$F_{(2.565, 91.39)} = 22.02$, $p < 0.0001$; significant interaction [$F_{(8, 285)} = 20.49$, $p < 0.0001$]; male: significant effect of lever [$F_{(1, 32)} = 92.39$, $p < 0.0001$; significant

effect of schedule [$F_{(2.532, 77.85)} = 32.38$, $p < 0.0001$]; significant interaction [$F_{(8, 246)} = 33.60$, $p < 0.0001$]). After training, a dose–response curve was run on an FR1 schedule of reinforcement to assess shifts in the reinforcing efficacy of cocaine between males and females (see Supplementary Fig. 3a, b). There was a significant difference in the dose–response function (Supplementary Fig. 3b), and female mice show a trend toward a shifted peak of their dose–response curve as compared to males (Supplementary Fig. 3c). Moreover, when mice responded for cocaine (0.3 mg/kg/injection)

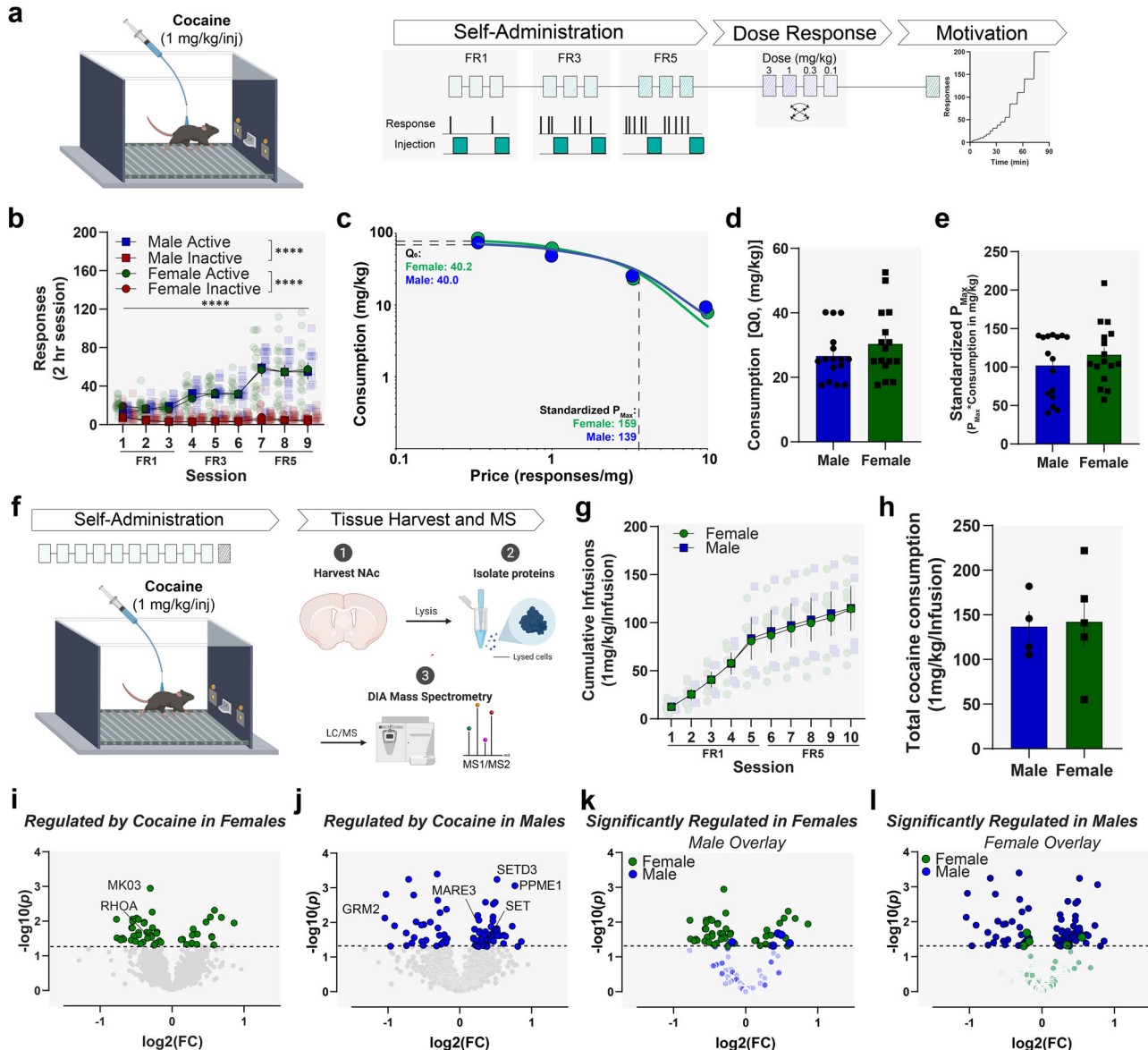

**Fig. 3 Cocaine self-administration regulates different proteins in males and females even when cocaine intake and self-administration behavior are not different. a** A series of behavioral experiments were run to assess sex differences in motivation for cocaine self-administration in males and females. Schematic/timeline of self-administration. **b** Average responses for cocaine under escalating fixed ratio schedules in male and female mice. Male and female mice acquire and consume cocaine at comparable rates under FR1, 3, and 5 schedules of reinforcement. **c** A concentration–response curve was run across days with doses counterbalanced between animals (0.1, 0.3, 1, and 3 mg/kg). Data were plotted as a demand curve where consumption was plotted on the $y$-axis and price (in responses required to obtain 1 mg cocaine) was plotted on the $x$-axis. Curves were fit to determine consumption at a minimally constraining price ($Q_0$) and the maximal price paid ($P_{max}$) in males and females. Standardized $P_{max}$ ($Q_0 \times P_{max}$) was calculated to allow for comparisons that are not influenced by the relative level of consumption and are comparable across groups. **d** $Q_0$—plotted as mg/kg to control for body weight differences—was not significantly different between males and females. **e** Standardized $P_{max}$ was also not significantly different between males and females. **f** In a separate group of animals, mice were trained to self-administer cocaine (or saline for control) for 10 days, after which NAc tissue was collected and processed for mass spectrometry. Schematic of self-administration, tissue collection, and processing for mass spectrometry. **g** Male and female mice consumed cocaine at the same rate and **h** there were no differences in total cocaine consumption. **i** Proteins significantly altered by cocaine self-administration in the NAc of female mice. Dotted lines on the volcano plot denote the significance cut off. **j** Proteins significantly altered by cocaine self-administration in the NAc of male mice. **k** Volcano plot showing only the proteins that are significantly regulated in females and those same proteins in males. Most of the proteins significantly regulated in females are not regulated in males. **l** Volcano plot showing only the proteins that are significantly regulated in males and those same proteins from the female group. All behavior data reported as mean ± SEM. *$p < 0.05$, ****$p < 0.0001$. Diagrams created with biorender.com.

under a progressive ratio schedule, female mice showed a trend towards a higher break point compared to males, potentially indicating a higher level of motivation at this dose (Supplementary Fig. 3d). However, the biggest differences in the dose–response relationship were at the lowest doses which could be influenced by factors other than motivation. Therefore, we analyzed these data

more rigorously and assessed a pure measure of motivation. To this end dose–effect functions were plotted as demand curves (Fig. 3c). We found that at a minimally constraining price, female mice consume ($Q_0$) comparable levels of cocaine as compared to male mice (Fig. 3d). Moreover, we find the maximal price female mice are willing to pay (standardized $P_{max}$) for cocaine is also not different

from male mice (Fig. 3d, e). These data are similar to data we have reported previously in rats, showing no sex differences in $P_{max}$ and $Q_0$ for cocaine, except under specific conditions[20].

**A history of cocaine self-administration induced nearly non-overlapping changes in protein expression between males and females.** As shown in Fig. 2, many of the sexually dimorphic proteins at baseline were linked specifically to reward and drug responses, thus, we wanted to understand how cocaine-induced plasticity in the proteome differed between males and females. Using the experiments above (also see Supplementary Fig. 3), we selected a dose of cocaine that resulted in similar cocaine intake and reinforcement behavior between males and females to ensure proteomic differences between the sexes were not due to differences in drug intake or response rates. Male and female mice self-administered cocaine (1 mg/kg/injection) or saline for 10 consecutive days. Twenty-four hours following the final self-administration session mice were sacrificed and NAc tissue was analyzed with DIA mass spectrometry (Fig. 3f). Importantly, the proteomics experiments to assess sex differences in animals without a cocaine history—noted above—were conducted by comparing the male and female subjects from the saline group, thus all animals were run in a single cohort. Male and female mice acquired cocaine self-administration at equal rates (Fig. 3g; two-way repeated measures ANOVA: main effect of session [$F_{(1.045, 8.363)} = 45.81$, $p = 0.0001$], no effect of sex [$F_{(1, 8)} = 0.009$, $p = 0.93$], no significant interaction [$F_{(9, 72)} = 0.030$, $p > 0.99$]) and consumed equal amounts of cocaine (Fig. 3h; unpaired two-tailed $t$ test: $t_7 = 0.16$, $p = 0.88$)—indicating that any subsequent proteomic differences are not due to differences in drug intake or reinforcement behavior. In female mice that had self-administered cocaine there were 50 significantly regulated proteins compared to saline controls (Fig. 3i and Supplementary Data 3). In males, 82 significantly regulated proteins were identified following cocaine self-administration compared to male saline controls (Fig. 3j and Supplementary Data 4). In both male and female mice, we identified dysregulation of proteins previously implicated in drug-associated behaviors (e.g. females: MK03[30], RHOA[31]; males: GRM2[22,32], MARE3[33], and PPME1[34] (Fig. 3i–l).

Importantly, we found that proteins regulated by cocaine self-administration are not modulated similarly in the other sex (Fig. 4a). For example, of the 50 proteins regulated by cocaine in females, many of them are not altered, or showed an opposite pattern of expression in males (see Fig. 4a, right). Overall, of the 127 total proteins regulated by cocaine self-administration in both males and females, only 5 are commonly regulated in the NAc of both sexes (Fig. 4b, left and see Supplementary Data 5). Within this population of commonly regulated proteins, two of these proteins were significantly regulated in the opposite direction in males and females (Fig. 4b, right). We conducted odds ratio analyses to determine the probability of overlap, and whether it differs from what would be predicted if there were no effect. Using this approach, we found there is a significant overlap between the proteins unchanged by cocaine in males and those unchanged by cocaine in females (compared to predicted overlap by random chance), while the overlap in proteins changed by cocaine in males and those changed by cocaine in females did not reach statistical significance (odds ratio = 2.37, $p = 0.06$, Fig. 4c). This method demonstrates statistically that cocaine is indeed altering nonoverlapping protein populations between male and female mice. Finally, to characterize this effect further, we used GO analysis to identify GO terms enriched in the NAc of either female (Fig. 4d) and male (Fig. 4e) mice after cocaine self-

administration. Many similar GO terms were identified in males and females, thus it is possible that males and females have similar cocaine-associated behavioral profiles being mediated by different molecular mechanisms[19].

**Prior cocaine exposure eliminates sex differences in the NAc proteome.** The above data show that proteomic regulation by a history of cocaine self-administration regulates nearly non-overlapping proteins between male and female mice. However, the next question was whether cocaine effects on the proteome were independent of baseline differences between males and females, or, whether cocaine-induced plasticity had differential effects that were influenced by the sexual dimorphisms at baseline. The power of this proteomics approach is that it assesses a large array of proteins between the groups, allowing for the identification of complex patterns across the proteome —something that would not emerge with single protein analyses.

To identify large-scale patterns that were induced by cocaine self-administration, we used rank–rank hypergeometric overlap (RRHO) comparisons, which allow for not only threshold-free identification of patterns that emerge across the entire proteome, but also comparisons between two differential lists to determine global similarities and differences, as well as relative relationship (Fig. 5a). The RRHO analysis was conducted using the Log2FC, meaning that the larger $p$ values presented in the RRHO output denote more significant overlap determined by relative effect sizes in differential protein expression at those points. Using the stratified RRHO2 method[35], we find concordance in protein expression patterns in male and female mice following cocaine self-administration, in relation to sex-specific saline controls (Fig. 5b; peak $-\log_{10}(p \text{ value}) = 12$)—suggesting that across the entire proteome there are similar patterns of cocaine regulation in the directionality and relative protein changes, even if these proteins do not reach statistical significance. However, when comparing the proteins that were sexually dimorphic at baseline to the effects on these specific proteins after cocaine self-administration a different and more robust pattern emerged. In both females (Fig. 5c; peak $-\log_{10}(p \text{ value}) = 42$) and males (Fig. 5d; peak $-\log_{10}(p \text{ value}) = 29$), baseline sexually dimorphic proteins were regulated by cocaine in a way that would eliminate these sexual dimorphisms. Specifically, we find that in both male and female proteomes, cocaine is increasing the relative expression patterns of proteins downregulated at baseline, while decreasing the relative expression patterns of proteins upregulated at baseline (Fig. 5c, d). Therefore, while there is some overlap in the proteome in general following a history of cocaine self-administration, the larger—and more significant—effect of cocaine is on eliminating these baseline sexual dimorphisms.

To further explore the effects of cocaine on baseline sex differences, we compared the proteins that were differentially expressed at baseline between males and females and following cocaine self-administration in each sex (Fig. 5e, left). We find that of the 112 sexually dimorphic proteins at baseline, only 7 continue to be sexually dimorphic following cocaine self-administration (Fig. 5e, right). Moreover, using both Venn and odds ratio comparisons, we find that proteins that are sexually dimorphic at baseline are more likely to be subsequently altered by cocaine in males and females, compared to respective saline controls (Fig. 5f, g). Specifically, we find a significant overlap in proteins that are sexually dimorphic at baseline and those changed by cocaine in female mice (odds ratio = 6.3, $p < 0.0001$) and those changed in male mice (odds ratio = 4.2, $p < 0.0001$). However, we find no significant overlap between proteins sexually

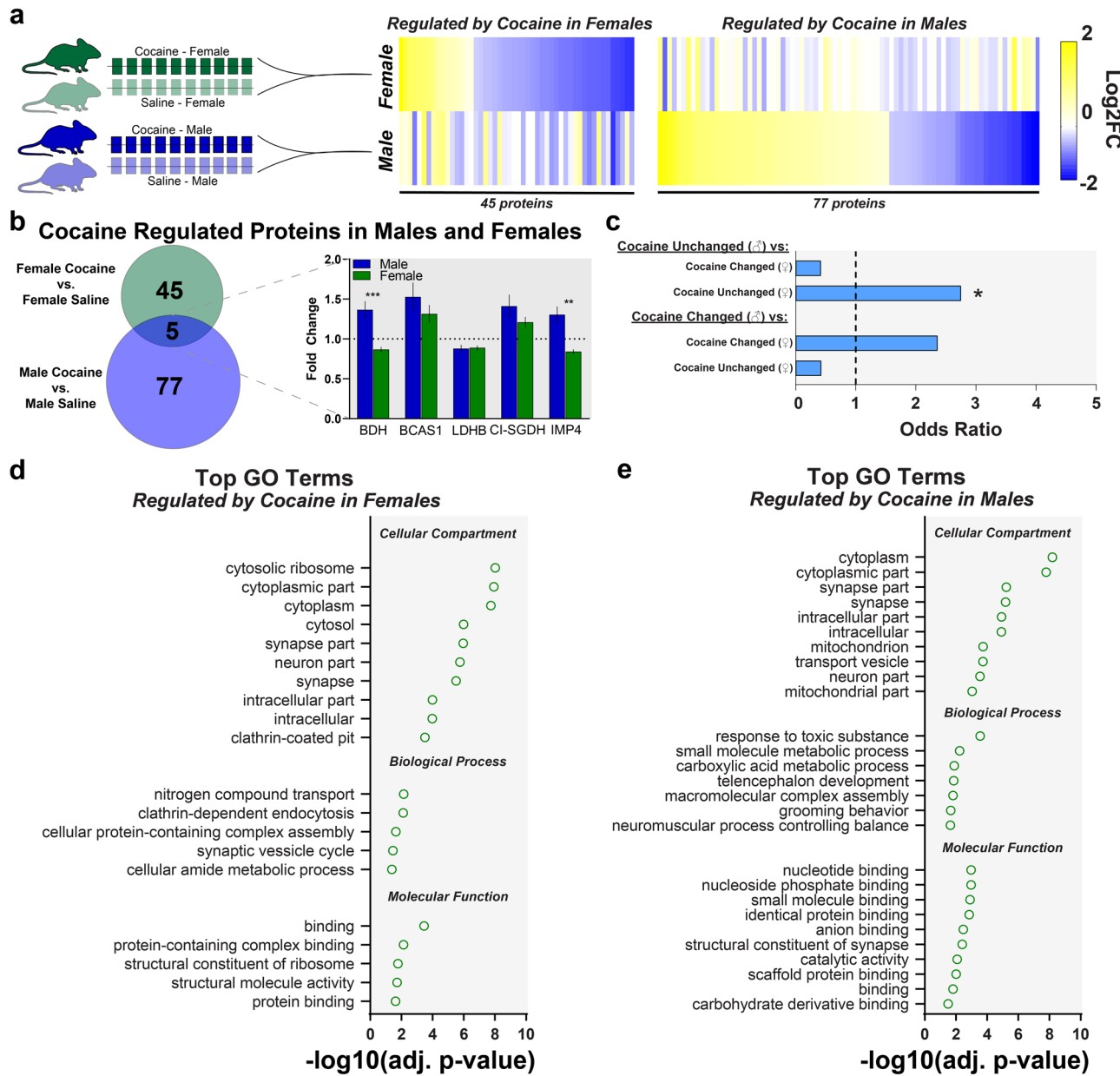

**Fig. 4 Cocaine-regulated proteins are largely nonoverlapping in the accumbens of male and female mice. a** Heat map denoting the proteomic dysregulation induced by cocaine. Each line is the differentially expressed proteins in the cocaine group in each sex compared to their respective saline controls. (Left) proteins significantly regulated by cocaine in females are on the top line. The bottom line is the same proteins in males (regardless of whether they were significantly different). (Right) the bottom line are the proteins significantly regulated by cocaine in males and the top line are those same proteins in females. **b** Of 122 total proteins regulated by cocaine self-administration in either males or females, only 5 were regulated in both sexes. (Inset) bar graph showing fold change in protein expression versus saline in each sex. Of the five proteins regulated in both sexes, only three of those were going in the same direction. **c** Fisher's exact test of odds ratio for protein list overlap of cocaine-regulated proteome in males and females. Odds ratio analysis shows that proteins regulated by cocaine in males are largely nonoverlapping with proteins regulated by cocaine in females. In order to conduct gene ontology (GO) analysis protein names had to be converted to associated gene names. These conversions are available in Supplementary Data 1. Top GO terms in proteins regulated by cocaine in **d** female and **e** male mice. Bar graphs reported as mean ± SEM. *$p < 0.05$, **$p < 0.01$, ***$p < 0.001$.

dimorphic at baseline and those unchanged by cocaine in female mice (odds ratio = 0.1, $p = 0.999$) and those unchanged in male mice (odds ratio = 0.2, $p = 0.999$; Fig. 5g, left). Conversely, we find a significant overlap between proteins sexually monomorphic at baseline and those unchanged by cocaine in female mice (odds ratio = 3.9, $p < 0.0001$) and those unchanged in male mice (odds ratio = 2.9, $p < 0.001$; Fig. 5g, right). These results demonstrate that cocaine is more likely to regulate these sexual dimorphic proteins than proteins that are monomorphic at baseline.

**Cocaine exposure eliminates sex differences in motivated behaviors**. These data suggest that despite seeing limited overlap in significantly regulated proteins between males and females, cocaine self-administration is neutralizing the baseline sex differences seen in the NAc prior to cocaine exposure. As mentioned above, the molecular profiles of the NAc are critical regulators of motivated behaviors and likely contribute to the sex differences that we and others have seen in naive animals. Here, we show that cocaine self-administration eliminates sex differences in the

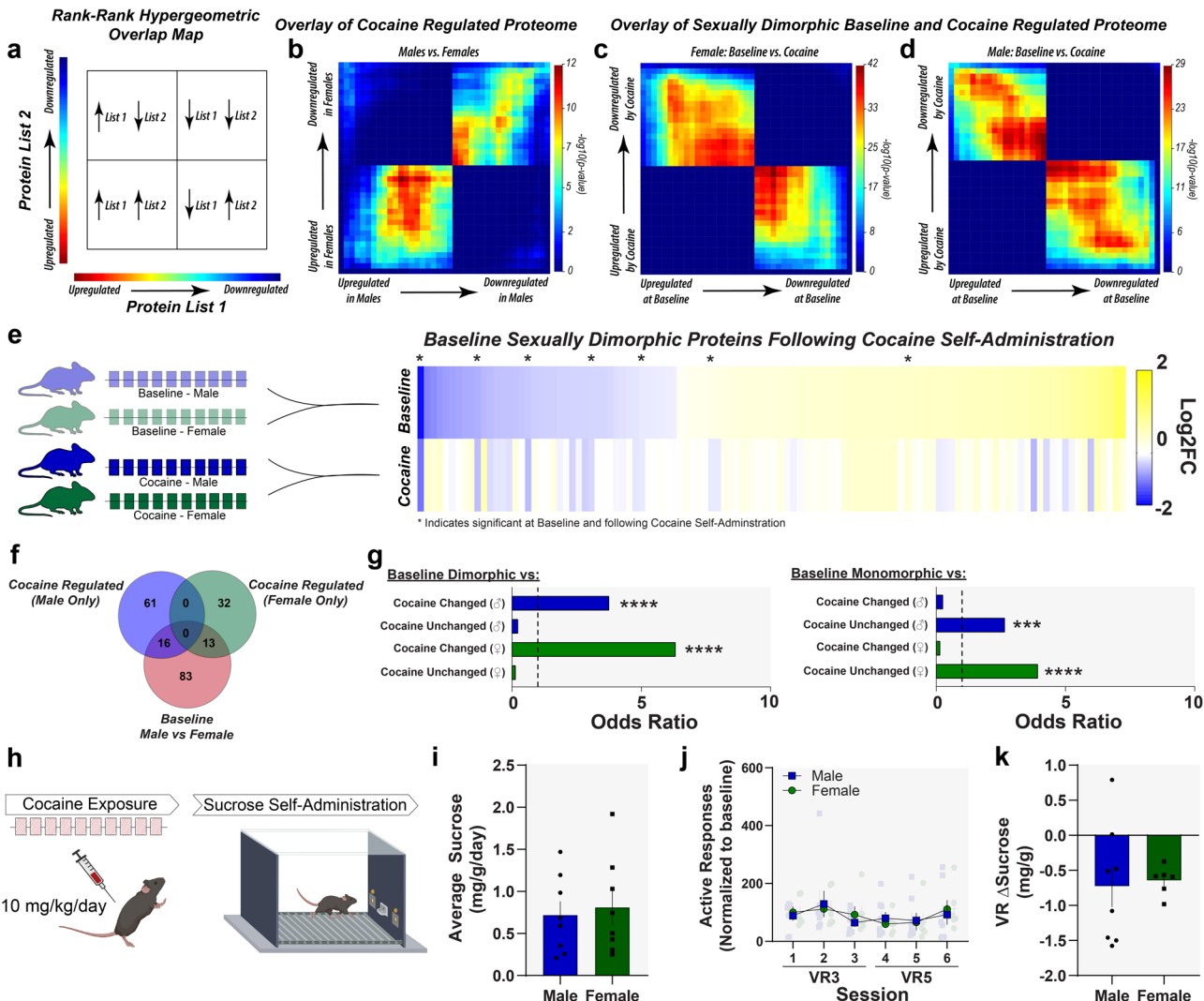

**Fig. 5 Cocaine exposure eliminates baseline sexual dimorphisms in the proteome and behavior. a** Rank–rank hypergeometric overlap (RRHO) plots allow for threshold-free comparisons in proteomic expression between two differential expression lists. **b** Cocaine-induced proteomic regulation (cocaine versus saline controls) in males and females was compared. There was some overlap between the cocaine-regulated proteins between males and females. **c** Cocaine had a much different regulation pattern for the baseline sexually dimorphic proteins than the rest of the proteome. RRHO plot showing the comparison between the proteins that were increased/decreased in females at baseline (x-axis) and the proteins following cocaine self-administration (female cocaine versus female saline). There was significant opposite regulation by cocaine in these proteins suggesting that cocaine increased downregulated and decreased upregulated sexual dimorphisms in females. **d** The same pattern was seen in males. RRHO plot showing cocaine-regulated proteins (cocaine versus saline controls) proteins that were increased/decreased in males at baseline. **f** Heat map comparing male and female proteomes at (top) baseline and (bottom) following cocaine. Of 112 proteins differentially expressed between males and females at baseline, only 7 (asterisk) are also significantly different between males and females following cocaine, suggesting that cocaine exposure eliminates these sexual dimorphisms. **f** Venn diagram of proteins regulated by cocaine in males (blue), regulated by cocaine in females (green), and differentially expressed between males and females at baseline (red). Proteins regulated by cocaine in both males and females appear to show enrichment in sexually dimorphic population at baseline. **g** (Left) Fisher's exact test of odds ratio for protein overlap of cocaine-regulated proteome in males and females compared to proteomic differences at baseline. **g** (Right) Fisher's exact test of odds ratio of cocaine-regulated proteome in males and females compared to sexually monomorphic proteins at baseline. Proteins that were sexually dimorphic at baseline were more likely to be regulated by cocaine than proteins that were not. **h** Lastly, we wanted to understand how cocaine exposure altered sexually dimorphic behaviors that are known to be controlled by the NAc. Therefore, we assessed how chronic cocaine pretreatment prior to sucrose self-administration altered behavior. **i** Average daily sucrose consumption in mg between male and female mice, normalized to body weight. There is no significant difference in sucrose consumption between males and females. **j** Average responses for sucrose under VR3 and VR5 between male and female mice, normalized to baseline (responses during first day of VR3), showing no sex differences. **k** Average change in sucrose consumption from VR3 to VR5, normalized to body weight. Male and female mice show comparable schedule-dependent changes in sucrose consumption, indicating that cocaine exposure eliminates sexually dimorphic behavior associated with nondrug reinforcers. All behavior data reported as mean ± SEM. *$p < 0.05$, ***$p < 0.001$, ****$p < 0.0001$. Diagrams created with biorender.com.

proteome at baseline; therefore, we hypothesized that cocaine exposure would eliminate sex differences in behavior for nondrug rewards as well.

To test this, male and female mice were injected with cocaine (10 mg/kg/day, i.p.) for 10 consecutive days (Fig. 5h, left).

Animals were subsequently trained to respond for sucrose (10 μL/delivery, 10% sucrose) on an FR1 schedule of reinforcement (Fig. 5h, right and see Fig. 1a). Following cocaine exposure, male and female mice consume similar levels of sucrose at baseline, showing that the sex differences observed before cocaine exposure

(shown in Fig. 1) are no longer present (Fig. 5i; unpaired two-tailed $t$ test: $t_{14} = 0.345$, $p = 0.74$). Furthermore, male and female mice respond for sucrose at similar rates under VR schedules of reinforcement (Fig. 5j; two-way repeated measures ANOVA: no effect of sex [$F_{(1,\ 12)} = 0.013$, $p = 0.91$), and there was no difference in effort at higher schedules of reinforcement between male and female mice (Fig. 5k; unpaired two-tailed $t$ test: $t_{12} = 0.23$, $p = 0.82$). These data suggest that while cocaine establishes unique proteomes in the NAc of male and female mice, repeated cocaine also eliminates baseline sex differences seen in reward-associated behavior.

## Discussion

While a large amount of work has focused on understanding how cocaine self-administration results in molecular plasticity in reward-related brain regions to drive the development of SUD, this work has overwhelmingly focused on male subjects. Here, we show that sex is a critical biological variable that defines drug-induced plasticity in the NAc. We assess the wide range expression patterns in addition to the individual protein targets altered by cocaine, and demonstrate unique behavioral and molecular profiles in male and female mice that are subsequently ablated following repeated cocaine exposure. We show that (1) prior cocaine exposure via intravenous self-administration induces different protein expression patterns (in significantly regulated targets) in males and females and (2) cocaine acts on baseline sexual dimorphisms to exert these effects. Critically, we find that cocaine administration blunts not only the previously identified sex differences in the accumbens proteome, but also the preexisting sex differences in behavior for nondrug reinforcers. Together, these data highlight the importance of understanding large-scale patterns of protein expression between males and females. These data also suggest that the molecular mechanisms and pathways engaged in males and females may not be the same, and even behaviors that look similar (i.e., cocaine reinforcement, sucrose reinforcement under certain schedules) may have different mechanistic bases. This underscores the importance of understanding the female-specific factors that contribute to SUD, in order to develop efficacious treatments in both sexes[36].

As others have previously demonstrated across a number of species[1,5,12,20,37], we show that females are more motivated to self-administer positive reinforcers in multiple tasks. These differences were much more robust for nondrug reinforcers (such as sucrose) than drug reinforcers (such as cocaine[12,20]). These differences are likely driven by differences in appetitive behaviors, rather than purely consummatory behaviors, as previously we did not observe differences in home cage feeding between males and females under similar conditions[12]. Further, while some measures of cocaine self-administration were enhanced in females, when we did behavioral economic analysis of these data we did not see significant differences between males and females, which is consistent with our previous work in rats[20]. Together, there were robust sex differences for sucrose, and subtle sex differences for cocaine, self-administration. Because of these differences in motivated behaviors, it is likely that there are significant molecular differences that exist in the NAc before drug is encountered that influence the way that drugs drive behavior, and can alter the plasticity that occurs after repeated drug exposure. Indeed, here we observe that these sex-specific behavioral differences were concomitant with baseline sex differences in reward-associated proteins in the NAc that have previously been linked to motivation, drug effects, and reinforcement behavior (such as GRM2 and CYFP2[26,38,39]).

When we assessed proteomic regulation following cocaine self-administration in males and females, the observed sex differences were not in the magnitude of protein expression, but rather different proteins were significantly regulated by a history of cocaine self-administration between males and females. First, our data recapitulates previous findings in males demonstrating cocaine-induced changes in protein expression in the NAc. Of note are LDHB, ACTB, NFL, GFAP, SNX27, and EF1G, which have been previously demonstrated to be regulated, following cocaine exposure in the NAc of humans and nonhuman primates[40–44]. We also identified several proteins previously shown to be disrupted in rodent striatum, including SNX27, CAD13, LMNA, CATB, MARE3, MK03, CBPE, SV2B, and RHOA[31,42,45–51]. MARE3 specifically has been shown to be regulated by cocaine self-administration and is a critical regulator of drug seeking that incubates over withdrawal[33]. In addition, we identified disrupted glutamatergic-specific protein expression (e.g., GRM2 and VGLU3), which are heavily implicated cocaine-induced adaptations in the NAc and cocaine-associated behaviors—especially as they relate to drug-induced plasticity in medium spiny neurons and glutamatergic inputs into them[22,25,38]. Thus, together these data, especially in males, are supported by a wide body of previous literature highlighting these same effectors.

However, when we compared the proteins regulated by prior cocaine self-administration in males to those that were regulated in females, there was nearly no overlap between the two protein lists. Of the 127 proteins regulated by cocaine in males and females, only five were regulated in both sexes and of those only three were regulated in the same direction. Importantly, these differences cannot be explained by differences in drug intake or behavior, as we specifically chose a dose of cocaine and a schedule of reinforcement that produced the same behavior in both sexes. This is also important as many sex differences are often expected to manifest as relative magnitude differences. For example, a protein may be regulated in both males and females, but exhibits a higher fold change in females. Here, we show that for cocaine-induced plasticity in the NAc that this is not the case. Rather, cocaine results in the significant regulation of different proteins in females. Together, these data suggest different mechanistic effects underlie similar behavioral changes between males and females, and highlight the critical importance of understanding the female-specific factors that underlie drug use and plasticity in females.

Importantly, these cocaine-induced differences in NAc protein expression were driven, at least in part, by preexisting sex differences in the NAc proteome. The different expression profiles between males and females seem to be largely driven by an interaction between basal protein expression and the effects of cocaine self-administration. In both male and female mice, proteins that were downregulated at baseline (as compared to the opposite sex) were increased by cocaine self-administration, while proteins that are upregulated at baseline are decreased by cocaine self-administration. It is possible that repeated cocaine exposure promotes cocaine-seeking by ablating the molecular signatures of other behavioral strategies—especially as they relate to nondrug reinforcers[12,20,52,53]. Indeed, following repeated cocaine exposure, male and female mice no longer show differences in basal sucrose consumption or motivation for sucrose. Although comparable behavioral outputs are observed, there are unique sex-specific molecular mechanisms altering NAc function during cocaine self-administration. Furthermore, while our proteomic analysis did identify individual protein targets significantly regulated in a sex-specific fashion, more complex patterns make up a large portion of sex differences in cocaine response. In addition to the individual protein dysregulation, our RRHO analyses demonstrate that cocaine also alters large-scale proteomic networks throughout the

NAc in a manner that dampens preexisting sex differences in protein expression.

These data highlight a critical aspect of sexual dimorphisms in the brain and behavior: there can be differences in the molecular and cellular mechanisms regulating similar behavioral profiles. Thus, a lack of sex differences in behavior are not demonstrative of a lack of sex differences in the mechanistic control of that behavior within the brain. Regarding these mechanisms, it is particularly important to note that basal levels of many of these proteins are present in the NAc of both male and female mice. As such, studies which manipulate these targets (commonly, through genetic knockout or viral overexpression) will likely identify changes in cocaine-associated behaviors for both male and female subjects, despite the fact that cocaine is only selectively engaging these proteins in a sex-specific manner. Accordingly, without considering sex as a biological variable, various targets with therapeutic potential in females will be unidentified or targets with minimal therapeutic potential will be selectively studied. Moving forward, it will be critical to parse individual gene targets regulated by drugs of abuse from the larger effects on gene networks, which ultimately dictate cellular function between the sexes.

## Methods

**Animals**. Eight-week-old male and female C57BL/6J mice were ordered from Jackson Laboratory (Bar Harbor, Maine). Twelve-week-old male and female Sprague Dawley rats were ordered from Envigo (Madison, WI). Mice were group-housed (4–5/cage) and rats were pair-housed under a 12:12 h reverse dark/light cycle with food and water ad libitum. Experiment-specific use of mice or rats is denoted in the methods below and appropriate results sections. During all self-administration experiments (except for the self-administration under sated conditions in Fig. 1b), animals were food restricted to 95% body weight (for mice) and 85% body weight (for rats). All behavioral experiments were conducted during the dark cycle. Experiments were approved by the Institutional Animal Care and Use Committee of Vanderbilt University Medical Center. All experiments were conducted according to the National Institutes of Health guidelines for animal care and use.

**Drugs**. Cocaine-HCl (NIDA Drug Supply Program) was dissolved in sterile saline (0.9% NaCl). Ketamine-HCl and xylazine (Patterson Veterinary) were mixed fresh in sterile saline (0.9% NaCl).

**Operant conditioning**. Male and female mice were trained to self-administer liquid sucrose (10% w/v), as previously described[12]. Briefly, mice were trained in standard mouse operant chambers (Med Associates, St. Albans, USA) equipped with two illuminated nose pokes, cue lights above each nose poke, and a white noise generator with speaker. For each task, one nose poke was designated as the "active poke" and the other as the "inactive poke". The schedules under which the reinforcer was delivered were changed depending on the experiment and are outlined below.

**Sucrose self-administration in mice**. Male ($n = 8$) and female ($n = 10$) mice responded for sucrose under a FR1 schedule of reinforcement over four consecutive daily sessions (2 h). Active responses resulted in a liquid sucrose delivery concurrent with the illumination of a cue light directly above the operanda for 5 s; inactive responses resulted in the presentation of a distinct light cue, but no sucrose delivery. Next, male ($n = 5$) and female ($n = 5$) mice were food restricted to ~95% body weight and were trained under FR1, VR2, VR3, and VR5 schedules of reinforcement (3 consecutive days of each schedule, 30 min/day). For FR1 each active lever press resulted in the delivery of sucrose. For VR schedules sucrose delivery occurred after an average of 2, 3, or 5, presses, but was variable across the session.

**Within-session threshold procedure**. Male ($n = 7$) and female ($n = 11$) rats were magazine trained for three sessions until reliably consuming 90 chocolate sucrose pellets within 60 min. Subsequently, rats were allowed to lever press to receive sucrose pellets under an FR1 schedule of reinforcement for 60 min. Once stable responding for sucrose pellets was reached, rats were tested in the within-session threshold procedure. Briefly, rats went through 11 10-min bins during which the animals could earn up to 20 sucrose pellets available upon a series of increasing FR schedules (1, 2, 3, 5, 8, 10, 14, 18, 25, 31, and 48) with no timeouts, and levers remain extended. Consequently, the "price" of sucrose—the number of responses required to obtain each pellet—increases over time. Both the number of lever

presses and the total intake are measured for each price bin and plotted as a function of price[20,54]. Data were plotted as demand curves and then analyzed, as described below.

**Behavioral economic analysis of demand curves**. The resulting demand curves can be used for mathematical curve fitting using the following equation: $\log_{10}(Q) = \log_{10}(Q_0) + k(e - \alpha Q_0 C - 1)$, where $Q$ is consumption and $C$ is the varying cost of the reinforcer[55]. The value $k$ was set to 2 for all animals[55]. The parameters $Q_0$ and $P_{max}$ were then calculated for each animal.

$Q_0$. $Q_0$ is a measure of the animals' preferred level of consumption. This is measured by determining the amount of reinforcer consumption when it is available at low effort, or a minimally constraining price.

$P_{max}$. $P_{max}$ is the price at which the animal no longer emits enough responses to maintain the $Q_0$ level of intake and consumption decreases. It is empirically defined as the inflection point of the demand curve and can be measured as the point for which the first derivative slope is equal to −1. As the session progresses, animals must increase responding on the active lever to maintain stable intake. $P_{max}$ is the price at which the animal no longer emits enough responses to maintain intake and consumption decreases. $P_{max}$ is expressed in units of $C$ (cost) in effort for a given reinforcer.

*Standardized* $P_{max}$. Is determined by the equation $P_{max}$ (in units of $C$) $\times Q_0$. Standardized $P_{max}$ controls for differences in intake that can influence $P_{max}$ and allow for a standardized measurement that can be compared across groups and species. This measurement is use throughout the manuscript unless otherwise noted.

Previous work has demonstrated that $P_{max}$ is highly correlated with break points on a progressive ratio schedule of reinforcement, confirming that the threshold procedure accurately assesses reinforcing efficacy and motivation[56].

**Jugular catheter implantation**. Mice were anesthetized with ketamine-HCl (100 mg/kg)/xylazine (10 mg/kg) i.p. and implanted with chronic indwelling jugular catheters, as previously described[57]. Ampicillin (0.5 mg/kg)/heparin (10 U/mL in sterile saline (0.9% NaCl) was administered i.v. daily. Mice recovered >3 days before commencing drug self-administration training.

**Cocaine self-administration**. Mice were trained to self-administer cocaine, as previously described[20,57]. Start of daily sessions (2 h) were signaled by initiation of white noise. A series of tasks were conducted to assess the reinforcing efficacy in males and females.

**Dose–effect function**. Male ($n = 16$) and female ($n = 17$) mice were trained under FR1, FR2, and FR5 schedules of reinforcement (1 mg/kg/injection, 3 days/schedule, 9 consecutive training days) to induce stable self-administration and confirm acquisition. Active responses resulted in a 3 s cocaine infusion concurrent with a presentation of the nose poke light for 5 s. Inactive responses had no programmed consequence. For dose–response testing, animals self-administered cocaine over 4 consecutive days at the following doses: 0.1, 0.3, 1, and 3 (mg/kg/injection, counterbalanced, FR1). Dose–response curves were plotted as demand curve functions and the parameters $Q_0$ and $P_{max}$ were then calculated for each animal, as described above.

**Motivational testing**. Subsequently, mice returned to FR1 at 1 mg/kg/injection until stable, and then subsequently tested under progressive ratio (PR; 0.3 mg/kg/injection), where the ratio began at 1 and increased with each reinforcer received; the list of ratios used was as follow: 1, 3, 5, 6, 8, 10, 14, 18, 25, 31, 38, 45, 55, 85, 110, 140, and 200. If no responses were recorded for 20 min, the final reached ratio was recorded, and the program terminated.

**Self-administration for mass spectrometry analysis**. Mice were trained to self-administer cocaine [1 mg/kg/injection, male ($n = 5$) and female ($n = 4$)] or saline (male $n = 5$, female $n = 5$) under FR1 schedule of reinforcement for 5 consecutive days and FR5 schedule of reinforcement for 5 consecutive days.

**Experimenter delivered cocaine**. Male ($n = 8$) and female ($n = 8$) C57BL6/J mice were injected with cocaine-HCl (10 mg/kg/day, i.p.) for 10 consecutive days in the home cage. Twenty-four hours following final cocaine injection, mice were trained in sucrose-reinforced operant conditioning, as previously described.

**Tissue collection and preparation for LC-MS/MS**. Twenty-four hours following cocaine self-administration, NAc tissue was harvested (1 mm × 14-gauge punch) and flash frozen on dry ice. NAc tissue was prepared for DIA MS, as previously described[54]. Briefly, NAc tissue was lysed in solubilization buffer (8 M urea and 0.4 mM ammonium bicarbonate, pH 8.0) using a probe sonicator. Cellular debris was pelleted by centrifugation and supernatant containing soluble proteins was

collected. Soluble protein (50 μg) was resuspended in solubilization buffer (volume to 100 μL) and reduced in dithithreitol (5 μL, 200 mM) at 37 °C for 30 min. Samples were alkylated in iodacetamide (5 μL, 500 mM) and incubated at room temperature protected from light for 30 min. Samples were further diluted in water to a final urea concentration of 2 M and incubated in sequencing-grade trypsin (1:20 trypsin: protein; Promega, Madison, WI, USA) at 37 °C for 16 h. Digested proteins were acidified in 0.1% formic acid and desalted via column purification (C18 spin columns; The Nest Group, Inc; Southborough, MA, USA) and dried using a rotary evaporator. Purified samples were resuspended in 0.2% trifluoroacetic acid (TFA)/2% acetonitrile (ACN) in water.

**DIA mass spectrometry.** Purified samples were resuspended in 0.2% TFA/2% ACN in water. DIA LC-MS/MS was carried out through a nanoACQUITY UPLC system (Waters Corporation, Milford, MA, USA) connected to an Orbitrap Fusion Tribrid (ThermoFisher Scientific, San Jose, CA, USA) mass spectrometer. Samples were injected and loaded into a trapping column (nanoACQUITY UPLC Symmetry C18 Trap column, 180 μM × 20 mm) at 5 μL/min. Peptides were subsequently separated using a C18 column (nanoACQUITY column Peptide BEH C18, 75 μm × 250 mm). Mobile phases consisted of mobile phase A (0.1% formic acid in water) or mobile phase B (0.1% formic acid in ACN). Peptides were eluted with 6–35% gradient mobile phase B for 90 min, 85% mobile phase B for 15 min at 300 nL/min, 37 °C. All sample injections were interspersed by column regeneration and three blank injections. Data were acquired under DIA mode with an isolation window of 25 $m/z$. Full scan was in the 400–1000 $m/z$ range, "use quadrupole isolation" enabled at an orbitrap resolution of 120,000 at 200 $m/z$ and automatic gain control target value $4 \times 10^5$. MS$^2$ fragment ions were generated in C-trap with higher-energy collision dissociation at 28% and orbitrap resolution of 60,000.

DIA spectra were searched against a *Mus musculus* brain proteome fractionated spectral library generated from DDA LC-MS/MS spectra (collected from the same Orbitrap Fusion mass spectrometer) using Scaffold DIA software v. 1.1.1 (Proteome Software, Portland, OR, USA). Within scaffold DIA, raw files were first converted to the mzML format using ProteoWizard v. 3.0.11748. The samples were then aligned by retention time and individually searched with a mass tolerance of 10 p.p.m. and a fragment mass tolerance of 10 p.p.m. The data acquisition type was set to "nonoverlapping DIA", and the maximum missed cleavages was set to 2. Fixed modifications included carbamidomethylation of cysteine residues (+57.02). Dynamic modifications included phosphorylation of serine, threonine, and tyrosine (+79.96), deamination of asparagine and glutamine (+0.98), oxidation of methionine and proline (+15.99), and acetylation of lysine (+42.01). Peptides with charge states between 2 and 4 and 6–30 amino acids in length were considered for quantitation, and the resulting peptides were filtered by Percolator v. 3.01 at a threshold FDR of 0.01. Peptide quantification was performed by EncyclopeDIA v. 0.6.12 (ref. [58]), and six of the highest quality fragment ions were selected for quantitation. Proteins containing redundant peptides were grouped to satisfy the principles of parsimony, and proteins were filtered at a threshold of two peptides per protein and an FDR of 1%.

**Proteomics analysis.** Proteins were excluded from analysis if they were not detected in >50% of all samples irrespective of treatment. Pairwise comparisons of the log10 median intensity of every remaining protein and protein group were made, using Scaffold DIA proteomics analysis software (http://www.proteomesoftware.com/products/dia/). Identified protein names were converted to gene names for subsequent network and functional analysis. See Supplementary Data 1 for the protein to gene name conversions with accession numbers. Proteins from NAc that were significantly regulated between males and females at baseline were uploaded into the STRING database (https://string-db.org/), and enrichment of protein–protein interactions was assessed using the "multiple proteins" query using default settings[58]. See Supplementary Data 6 for the list of proteins identified in the network. For visualization of protein–protein interaction networks disconnected nodes were removed from the image, and node size was adjusted to correspond to the −log(p) relative to control. Significantly upregulated and downregulated proteins from each brain region were separately uploaded into the open source GO pathway analysis software package G:Profiler (https://biit.cs.ut.ee/gprofiler/gost) to identify significantly enriched GOs and KEGG pathways, using an FDR corrected $p < 0.05$ (ref. [59], Supplementary Data 7–14). Related pathway terms were reduced using Revigo (http://revigo.irb.hr/; Supplementary Data 9, 10, 13, and 14)[60]. If five terms or less in a given category were identified using G:Profiler, these were not reduced further using Revigo and are reported in full in the figures. Given that these software utilize gene names for identifying pathways, all protein names were converted to gene names prior to pathway analysis using the Uniprot database (https://www.uniprot.org/uploadlists/; see Supplementary Data 1). Odds ratios and Fisher's exact test were calculated and plotted using the GeneOverlap R-Script[61]. RRHO plots were generated using the RRHO2 R-Script[35] (Supplementary Data 15–18). Protein names are reported as the UNIPROT official short name or UNIPROT entry designation for consistency throughout the study. We have also included a supplementary table (Supplementary Data 1), which includes the protein name, accession number, and gene name to ensure accurate identification of targets. Data are available via Proteome X change with identifier PXD022832 (ref. [62]).

**Proteomics comparator groups.** Proteomic analysis was done on mice that had underdone cocaine self-administration (described above, $n = 5$ males, $n = 5$ females) or saline self-administration as controls (described above, $n = 5$ males, $n = 5$ females). Baseline sex differences from Fig. 2 are from the male versus female saline control groups. Proteomic results from cocaine male and cocaine female groups are represented as fold change from sex-specific saline controls.

**RNA isolation and qRT-PCR.** Twenty-four hours following cocaine self-administration, mice were sacrificed, and brains were flash frozen on dry ice-chilled isopentane. Flash-frozen 1.0 mm$^2$ NAc tissue punches were collected from 3 × 300 μM coronal sections. RNA was purified from tissue using the RNeasy Mini kit (catalog #74106; Qiagen) and total RNA was reverse transcribed using High-Capacity cDNA Reverse Transcription Kit (catalog #4368813; ThermoFisher Scientific). cDNA was analyzed through SYBR green chemistry (iTaq Universal SYBR Green Supermix) via BioRad CFX96 and CFX Maestro software. All values were calculated via the Pfaffl method and normalized to *Hprt1* expression generated simultaneously. Primers were generated using IDT PrimerQuest; *Hprt1* left: 5′-CTTTGCTGACCTGCTGGATT-3′, *Hprt1* right: 5′-TATGTCCCCCGTTGACTGAT-3′; *Ddx3x* left: 5′-TGCTGGCCTAGACCTGAACT-3′, *Ddx3x* right 5′-GGTGGGATATAACGCCCTTT-3′; *Ddx3y* left: 5′-TTGTGCTCAAACAGGGTCTG-3′, *Ddx3y* right 5′-TTACGGCGACCATATCTTCC-3′; and *Cyfip2* left: 5′-CTGCACCCCTTCCAAAACATT-3′, *Cyfip2* right: 5′-CTATCAGGGGAACCAGGTGA-3′.

**Statistics and data.** Type I error rate (alpha) was set to 0.05 for all statistical tests, unless otherwise noted. Data are represented as the mean ± SEM in figures where appropriate. Data were analyzed and graphed using Graphpad Prism 8.2 (La Jolla, CA). Group comparisons of one independent variable were tested using a two-tailed Student's $t$ test. Group comparisons of two independent variables were tested using two-way ANOVA, with a Sidak post hoc analysis. All other data are available from the corresponding author on reasonable request.

**Reporting summary.** Further information on research design is available in the Nature Research Reporting Summary linked to this article.

## Data availability
The data that support these findings are available at ProteomeXchange via PRIDE with the identifier PXD022832.

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

## Acknowledgements

This work was supported by the NIH (DA042111 and DA048931 to E.S.C.; DA044308, DA-049568, and DA-051551 to D.D.K.; DA041838 to A.J.L.; DA047777 to A.R.J.; and T32MH064913 to K.C.T.), as well as funds from the Brain and Behavior Research

Foundation in the form of a Young Investigator Grant (to R.S.H., E.S.C., and D.D.K.). the Whitehall Foundation (to ESC), and the Edward Mallinckrodt Jr. Foundation (to E.S.C.). The Orbitrap Fusion mass spectrometer and the Offline UPLC utilized were supported in part by NIH SIG grants 1S10OD019967-0 and 1S10ODOD018034-01, respectively, and Yale School of Medicine. This work was supported by the Yale/NIDA Neuroproteomics Centre Grant DA018343. We would like to thank the NIDA drug supply program for providing cocaine-HCl used within this study. Portions of Figs. 1, 2, and 5 were created with biorender.com.

## Author contributions

A.J.L. and A.R.J. contributed equally to this manuscript. A.J.L., A.R.J., T.H.E., R.W., S.O.N., L.J.B., K.C.T., S.J.K., V.K., P.M., M.G.K., E.S.C., and T.T.L. conducted experiments. A.J.L., A.R.J., D.D.K., and E.S.C. generated experimental design and analyzed data. A.J.L., A.R.J., and E.S.C. designed figures and drafted manuscript.

## Competing interests

The authors declare no competing interests.
