## [Peer Review File · Communications Biology]

Reviewers' comments:

Reviewer #1 (Remarks to the Author):

NatComm-6926_0_Cocaine self-administration induces divergent protein expression in the nucleus accumbens of male and female mice to eliminate basal sex differences.

Alberto J. López, Amy R. Johnson Drew D. Kiraly, Erin S. Calipari

Females, whether human or experimental rodents, are more responsive to psychostimulants than males, whether self-administered or injected by an experimenter, as established by at least three experimental groups (Becker, Frick, McCarthy). The data in this paper agree, and then go on to perform proteomic analyses on the nucleus accumbens from the mice and rats. The interesting conclusion from the proteomics is that the sets of cocaine-responsive proteins in males and females are nearly non-overlapping, and that cocaine exposure pretty much always makes the sex-based differences smaller or disappear. The data are generally well presented and explained, with some points of clarification needed.

In Fig. 2 B and E, about half the proteins listed do not appear in the NCBI listing, only as Omniprot; the Methods sound as if NCBI gene names are being presented in the paper, since those are used for GO analyses, but clearly the names presented are Omniprot, so a major clarification on nomenclature is needed. Pad12 appears in neither Omniprot nor NCBI. Then there are proteins which are basically like “controls”, e.g. Ddx3y, which is only expressed in male tissues under any circumstance. It cannot be cocaine responsive in females.

Line 107 “hypothesis-free approaches” used to be denigrated if not demolished as “fishing”. It is not clear this is a wise comment to include.

Lines 228-229 “Using odds....no effect.” is not a sentence. Line 546 “Jose” should be capitalized. Line 546 has messed up punctuation. Reference 5 is mis-formatted.

Reviewer #2 (Remarks to the Author):

Lopez and Johnson et al., investigated sex differences in natural reward and cocaine self-administration in rodents as well as conducted proteomic analysis of the nucleus accumbens at baseline or after cocaine self-administration. They found that female mice and rats have a higher motivation for sucrose compared to males and a trend for higher motivation for cocaine. There are also sex differences in the proteome at baseline and after cocaine self-administration. However, when examined without threshold cut offs, cocaine self-administration largely has concordant proteome changes but discordant proteome alterations in cocaine self-administration compared to baseline in both sexes. Overall the results are very interesting and a step forward in understanding differences between sexes in the nucleus

accumbens and would be of interest to a wide range of readers. The paper may influence thinking in the field, particularly proteomic analyses that utilize all detected proteins rather than only significantly regulated proteins such as in the RRHO analysis. The statistical analyses utilized seem to be appropriate for the data but more information is needed to assess for some results. Overall, clarifications throughout the manuscript are necessary prior to publication but no further experiments need to be added. My concerns are as follows in no particular order:

1. A large source of confusion in the manuscript and data stem from switching between gene and protein identifiers between the manuscript, supplemental data, and figures. Additionally, some proteins are not identified correctly in the manuscript. The supplemental data tables with uniprot identifiers should also include gene names used for GO analysis and full protein names. SCL1A4 is identified in the manuscript but SATT in the figure, SLC17A3 should be SLC17A8 and the uniprot is VGLU3, VAMP1/synaptobrevin-1 in the manuscript is labeled as VAMP2 in the data and figure, CYFIP2 is CYFP2 in data, MAPK3 is MKO3 in data table and possibly MKO3 in figure, EEF1G is EF1G in supplement, CHD13 is CDH13 and CAD13 in data, DNM2 to DYN2, CTSB to CATB, EB3 to MARE3, CPE is difficult to identify in supplement. For proteins specifically highlighted in the manuscript it would be helpful to provide the full names.

2. The MS/MS raw data/proteomic data should be uploaded to a freely available database such as ProteomeXchange.

3. Full proteomics data should be provided as supplement such as ranked lists and p-values used for RRHO analysis, full GO data including member proteins for each GO term and proteins significantly regulated in analyses. This would especially help with interpretation of Fig 4F, were all significant GO terms included in Fig 4D and 4E which are summarized in Fig4F? Additionally, was the RRHO analysis conducted with the p-values? Please indicate in methods. A little more explanation of the differences in interpretation between the four quadrants for the RRHO analysis would aid in the interpretation of Fig5. Authors conclude that larger single peak p-values indicate a larger effect size, is this typically used with the RRHO method?

4. More details for the network analysis are important for clarity. What is the rationale for the STRING network analysis? Was the network in Fig 2F only created with significantly regulated proteins? It seems as though 2G and 2H are proteins only from the protein binding GO term and neuron projection term overlaid on the same network in 2F. This is not clear in the manuscript. Were these GO terms significantly overrepresented in the dataset? The manuscript mentions histone proteins and VAMP2/CYFP2/ACTB but not protein binding or neuron projection GO terms for these figures, are these hubs statistically significant or interesting highlights from the authors? Figures 2F, 2G, and 2H are difficult to read the individual protein names, particularly with more negative log fold changes (dark blue symbols).

5. Authors should clarify at the beginning of the results that mice are used throughout the study except for the behavioral economics analysis in Fig 1 and include a brief rationale for the use of rats in the

results section prior to those data.

6. These authors have previously published with behavioral economics, however additional methodological details are necessary to interpret this portion of the study. How are alpha and Q0 calculated for each animal? The methods section under “within session threshold procedure” should indicate rats in the title and also states Q0 is a measure of cocaine consumption rather than sucrose.

7. For the results of the rat behavioral economics experiment, why are Pmax for representative male and female rats in Fig1F and 1G beyond the possible price available to the rats and beyond the x axis? If 11 bins were used, why are 8 data points shown in these figures? Is Q0 statistically different between male and female rats? For clarity it would help to add a line for Q0 and a line for Pmax in the representative curves.

8. Methods section states that animals are food restricted during self-administration- is this sucrose or cocaine SA? The heading “operant conditioning” only includes information for mice, does this information also apply to the rats?

9. The introduction states that there is “almost non-overlapping molecular plasticity induced by cocaine self-administration between males and females” and the discussion states that “cocaine self-administration induces non-overlapping protein expression patterns in males and females”, however the results show that there is minimal overlap in individual statistically significant proteins, but concordant effects when examining the proteomics overall without arbitrary cutoffs. This is a strength of this manuscript and the introduction and discussion should reflect these results. The discussion also states that the RRHO analysis shows that cocaine alters the proteome in the NAc but does so differently in males and females, this seems opposite from the results.

10. The results state that “using odds ratio analyses to determine the probability of overlap and whether it differs from what would be predicted if there were no effect” appears to be a fragmented sentence. Also, “it is possible that males and females have similar cocaine-associated behavioral profiles being mediated different molecular mechanisms” also seems to be fragmented.

11. Do the odds ratio analyses use all detected proteins or only significantly altered proteins? Full statistics should be included for all of the odds ratio analyses.

12. Are there other interpretations of the final experiment where cocaine administration is blocking the sex differences in sucrose consumption? Were there any differences in locomotor activity after cocaine i.p. compared to the previous sucrose consumption experiment?

13. The discussion states that of the five overlapping proteins regulated by cocaine, only two are regulated in the same direction, but results show 3.

14. In the description for Fig3, panel D states that female mice show higher rate of behavior at higher

cocaine dose than males but results state this is a trend.

15. Fig4 panel B legend has total twice in a row. Methods section motivational testing paragraph has “until stable and then and subsequently”. DTT should be dithiothreitol and IAM should be iodoacetamide.

Reviewer #3 (Remarks to the Author):

This study examined protein expression patterns using large-scale proteomic analysis of the nucleus accumbens (NAc) from one set of naïve male and female mice and one set of male and female mice 24-h after a 10-d cocaine self-administration regimen. Behaviorally, there were no robust sex differences in cocaine self-administration behavior. However, in a sucrose self-administration paradigm, cocaine-naïve female mice (and rats) showed greater motivation-like behavior for sucrose compared to cocaine-naïve male mice (and rats). The proteomic data showed significant sex differences in the regulation of specific proteins and protein pathways in both naïve and cocaine-exposed mice – although GO analysis showed that cocaine regulates proteins with similar functions in male and female NAc. Not only were different proteins regulated in a sex-dependent manner, but the directionality of many proteins was different (e.g., upregulated in males and downregulated in females). The crux of the paper is the finding that cocaine-induced changes in protein regulation was such that it normalized male vs female protein levels under naïve conditions. In other words, if a protein was significantly reduced in male NAc compared to female NAc in naïve mice, that protein was likely to be upregulated in cocaine-exposed males (thus normalizing that protein to female levels). A partial test of this finding was done in which separate male and female mice were treated non-contingently with cocaine and then tested for motivation to self-administer sucrose. Whereas naïve female mice show greater motivation to self-administer sucrose, this sex difference was eliminated after cocaine injections. Although the paradigms are not exact, these data are consistent with the idea that cocaine-induced normalization of protein expression in the NAc also equalizes male and female sucrose self-administration.

This paper has many strengths, and there is a great deal of important, useful, novel, and potentially significant data. The combination of techniques and the use of powerful bioinformatic methods to address the question of how cocaine alters NAc plasticity and motivated behaviors are strengths. However, some of the strengths are weakened by assumptions of what the data represent, which is partly due to a lack of all necessary controls and partly due to inaccurate use of language to describe the results. This major issue and several other comments are described below.

1. In the Introduction (and elsewhere), clinical and preclinical findings specific to psychostimulants are generalized to all drugs of abuse. For example, within lines 79-83, it is stated “...when drug use is taken into consideration, female use exceeds that of males...” This is fairly specific to psychostimulants, and that should be made clear. A common and unfortunate consequence of generalizing is that incorrect assumptions are made by readers. Also, in the Discussion (lines 330 – 332), it is stated “...increased

sensitivity to drugs of abuse seen in females across species.” Please reword these types of generalizations to be more accurate.

2. Sucrose is used as a reinforcer and is described as a “natural reward”. I disagree that it is a natural reward, and this has been discussed in much of S. Ahmed’s work (see Lenoir et al., PLoS One, 2007). It doesn’t change the value of testing sucrose reward in the current study, but simply refer to it as a “reward” or a “reinforcer” or something similar.

3. Given #2, it isn’t accurate to refer to behavior in response to sucrose reward as “baseline”. Rather, there is sucrose self-administration behavior and cocaine self-administration behavior. These can be compared – but not as cocaine vs baseline reinforcement.

4. Throughout the manuscript, there are examples in which the language overstates the current findings or findings from other groups. For example, in lines 120-121, the statement “...characterize the molecular mechanisms regulating sex-specific reward behaviors in male and female mice” overstates what this paper does. There is no direct evidence presented that a particular molecular mechanism regulates sex-specific reward behaviors. There is indirect evidence.

5. I do not see the number of animals used for each study/treatment group written anywhere. Please add.

6. In Figure 1, graph D and its interpretation are confusing. If D is the change in sucrose from VR3 to VR5, and males and females had similar active responses in graph C, then it isn’t clear how there could be such a difference in graph D. The text (line 137) that males trended towards decreased consumption from VR3 to VR5. Again, I don’t see how that is possible given graph C.

7. Related to #6, the conclusion is made that females are “less sensitive to changes in cost than males”. But if the males are not really changing their consumption from VR3 to VR5 whereas females are upping their consumption, isn’t it more accurate to say that females are MORE sensitive to changes in cost, and adjust their behavior accordingly? Alternatively, males and females could be similarly sensitive, but males respond by reducing effort and females respond by increasing effort. I might not understand the concept of sensitivity in this context – some explanation is warranted.

8. A major weakness, alluded to in the beginning of this review, is that the different cohorts of mice are treated in different ways, and yet direct comparisons are made between the groups. Specifically, proteomic analysis was done on NAc tissue from naïve male and female mice (referred to as “baseline”) and on male and female mice that underwent cocaine self-administration and 24-h withdrawal. Results from those two cohorts were compared, and differences attributed to cocaine self-administration. But there are several other differences in the groups: naïve vs experience/self-admin behavior in the operant chambers, no surgery vs catheter surgery, minimal handling vs daily handling. Given these differences in the cohorts, there is rigor and validity to the reported sex differences in proteomics of (1) naïve mice and (2) mice that self-administered cocaine and had 24-h of withdrawal. However, it is not rigorous enough to conclude that the changes in protein expression after cocaine self-administration “eliminate” those differences that exist in naïve mice. Given that this is the primary conclusion of the paper, either an appropriate control group needs to be run (ideally mice that self-administer saline under the same conditions as the cocaine group) and the results compared to the cocaine self-admin group, or the conclusions of the paper need to be changed to reflect this issue.

9. Similarly, in the proteomic study from mice that underwent cocaine self-administration and 24-h withdrawal, it is not accurate to assume the protein changes are due to cocaine self-administration.

Changes in protein expression and plasticity could also arise from acute cocaine effects from the last cocaine self-admin session or from withdrawal effects. In this case, changing the text throughout the manuscript will suffice. It should not be stated that any proteomic changes are due to cocaine self-administration. It is ok to say “prior cocaine exposure” or “cocaine self-administration and 1 day withdrawal”, etc.

10. The behavioral experiment shown in Figure 5 is based on the idea that cocaine self-administration “eliminates” baseline sex differences in proteomics in the NAc, and this in turn eliminates sex differences in reward-seeking. This is supported by the finding that prior cocaine treatment results in males and females showing similar sucrose-seeking behavior (as opposed to Figure 1). Again, there is an issue with consistency of methods. The entire paper focuses on the effects of cocaine self-administration on protein expression, and yet the experiment in Figure 5 uses non-contingent cocaine injections.

11. Taking points 8-10 together, the line of reasoning and conclusions of the paper are flawed as written. They might be totally accurate, which would be fantastic. But the experiments were not conducted in a way that allows for those conclusions to be made.

12. Lines 229 – 232: The statement “...we found there is a significant overlap between the proteins unchanged by cocaine in males and those unchanged by cocaine in females (compared to predicted overlap by random chance), but no significant overlap in proteins changed by cocaine in males and those changed by cocaine in females (Fig. 4C)” isn’t strongly supported by the odds ratio data. Even though there was no significant overlap in proteins changed by cocaine in males and those changed by cocaine in females, the odds ratios for that comparison and the comparison of proteins unchanged by cocaine in males and those unchanged by cocaine in females are almost the same (~2.4 vs ~2.8). Some mention and discussion of this is warranted.

13. There needs to be validation of at least some of the protein changes determined with the mass spec. It would suffice to choose one or two proteins that are robustly up or down regulated in each sex/group and validate using western blots or some other method. As written, the entire proteomic data sets appear to come from one cohort of naïve mice and one cohort of mice that self-administered cocaine. This is not sufficient and does not instill confidence that the results would be replicable.

14. Please clarify if the cocaine dose-response experiment was initiated after the mice were done with the FR5 sessions at cocaine (1 mg/kg/infusion). It is advised that there be an extinction period after the 1 mg/kg self-admin period followed by demonstration that the mice can differentiate between vehicle and cocaine. This gives confidence in the dose response function. As it is displayed in Figure 3C, how do we know that the high number of active responses for the low 0.1 mg/kg dose is not just the mice seeking drug because they are used to getting 1.0 mg/kg and now they are suddenly getting 1/10th that dose?

COMMSBIO-20-3310-T

We would like to thank all reviewers for their constructive feedback on our manuscript. We are grateful for the enthusiasm, positive and supportive comments, and for their insightful suggestions for improvement. These comments were incredibly helpful in guiding the additional experiments, analysis and text edits, which we believe has considerably strengthened the manuscript. We also thank the reviewers for their patience throughout the revision process, as some of the comments required complex approaches to address. Together, we have been working tirelessly to include additional validation experiments and hope the reviewers will be pleased with the revised manuscript.

While all three reviewers expressed enthusiasm for the manuscript there were several comments that were brought up to enhance its impact. Thus, we have performed a **number of new experiments** and **additional analyses** to directly address these comments and concerns:

1. Reviewers 1-3 noted that it would be helpful to add more supplementary tables denoting gene-protein conversions and using consistent labeling to identify proteins throughout the manuscript. We have used the UniProt identifiers throughout the figures, tables, and manuscript and have included full protein names, accession numbers, and gene names that correspond to these labels in **Supplementary Table 1**.
2. Reviewer 2 noted that these data should be uploaded and freely available to the scientific community. The raw data has been uploaded to PRIDE and will be available upon publication.
3. Reviewer 2 had comments about determining differences in sucrose consumption between males and females in the operant tasks. We now include additional data showing that we do not see differences in consumption on low effort reinforcement schedules between males and females (**Supplementary Figure 1**)
4. Reviewer 3 had important suggestions for plotting cocaine self-administration data. We converted self-administration data into demand curves to represent sex differences more clearly (**Figure 3C-E, Supplementary Figure 3**).
5. Reviewer 3 requested an additional approach to further confirm the observations within the current study - we now include qPCR data for several targets (**Supplementary Figure 2**).
6. Reviewers 1-3 asked for additional supplementary data tables which are now included as **Supplementary Tables 1 - 18**
7. Reviewers 1-3 had suggestions for improving precision, accuracy, and clarity within the manuscript and figures. We have made edits to the figures and manuscript to address these concerns.

Changes within the main manuscript text are noted in **blue**. Please see below our point-by-point answer to the comments and concerns raised by the Reviewers.

Responses to Individual Reviewer Comments:

Please find point-by-point responses to each Reviewer's comments (*in italic and color-coded font*), with our responses below (in plain font).

Responses to Reviewer #1: Pages 2

Responses to Reviewer #2: Pages 3-7

Responses to Reviewer #3: Pages 8-14

References for entire Rebuttal: Pages 15

Reviewer #1:

1. Females, whether human or experimental rodents, are more responsive to psychostimulants than males, whether self-administered or injected by an experimenter, as established by at least three experimental groups (Becker, Frick, McCarthy). The data in this paper agree, and then go on to perform proteomic analyses on the nucleus accumbens from the mice and rats. The interesting conclusion from the proteomics is that the sets of cocaine-responsive proteins in males and females are nearly non-overlapping, and that cocaine exposure pretty much always makes the sex-based differences smaller or disappear. The data are generally well presented and explained, with some points of clarification needed.

Authors' Response: We would like to thank the reviewer for the positive comments, as well as the comments below that were critical in improving the clarity and accuracy of the manuscript.

2. In Fig. 2 B and E, about half the proteins listed do not appear in the NCBI listing, only as Omniprot; the Methods sound as if NCBI gene names are being presented in the paper, since those are used for GO analyses, but clearly the names presented are Omniprot, so a major clarification on nomenclature is needed. Pad12 appears in neither Omniprot nor NCBI. Then there are proteins which are basically like "controls", e.g. Ddx3y, which is only expressed in male tissues under any circumstance. It cannot be cocaine responsive in females.

Authors' Response: We thank the reviewer for bringing this up and prompting us to expand our clarification in the manuscript. First, we have carefully gone through the protein labels in the manuscript and figures. To ensure consistency we have used the UniProt Identifier in all places to label proteins.

For some of the computational tools that were used, we needed to convert the protein names to gene names. We have made changes to the methods and results to clarify when this occurred. We have also included a supplementary table that has the UNIPROT ID, accession number, full protein names, and the associated gene names that were used for any analyses where this conversion was necessary (**Supplementary Table 1**).

Further, we have specifically noted within the manuscript that some of the identified targets are only expressed in males or females, and as such their sex-specific expression profile is not due to cocaine-specific regulation. This is critical for presenting the data clearly and within the framework of cocaine-regulated expression and as such we agree with the reviewer that this is very important.

3. Line 107 "hypothesis-free approaches" used to be denigrated if not demolished as "fishing". It is not clear this is a wise comment to include.

Authors' Response: We agree and have changed the phrasing to say, "a data driven approach".

4. Lines 228-229 "Using odds....no effect." is not a sentence. Line 546 "Jose" should be capitalized. Line 546 has messed up punctuation. Reference 5 is mis-formatted.

Authors' Response: We have corrected these issues in the revised manuscript and have more thoroughly proofread for further potential errors.

Reviewer #2:

1. Lopez and Johnson et al., investigated sex differences in natural reward and cocaine self-administration in rodents as well as conducted proteomic analysis of the nucleus accumbens at baseline or after cocaine self-administration. They found that female mice and rats have a higher motivation for sucrose compared to males and a trend for higher motivation for cocaine. There are also sex differences in the proteome at baseline and after cocaine self-administration. However, when examined without threshold cut offs, cocaine self-administration largely has concordant proteome changes but discordant proteome alterations in cocaine self-administration compared to baseline in both sexes.

Authors' Response: We want to thank the reviewer for this accurate summary of our manuscript.

2. Overall the results are very interesting and a step forward in understanding differences between sexes in the nucleus accumbens and would be of interest to a wide range of readers.

Authors' Response: Thank you for the positive comments about the importance of these studies.

3. The paper may influence thinking in the field, particularly proteomic analyses that utilize all detected proteins rather than only significantly regulated proteins such as in the RRHO analysis. The statistical analyses utilized seem to be appropriate for the data but more information is needed to assess for some results. Overall, clarifications throughout the manuscript are necessary prior to publication but no further experiments need to be added.

Authors' Response: We want to thank the reviewer again for their support for the findings and their impact, but also for the critical comments and suggestions that have guided our revision. The changes suggested by the reviewer have greatly strengthened the clarity and impact of the manuscript.

4. A large source of confusion in the manuscript and data stem from switching between gene and protein identifiers between the manuscript, supplemental data, and figures. Additionally, some proteins are not identified correctly in the manuscript. The supplemental data tables with uniprot identifiers should also include gene names used for GO analysis and full protein names. SCL1A4 is identified in the manuscript but SATT in the figure, SLC17A3 should be SLC17A8 and the uniprot is VGLU3, VAMP1/synaptobrevin-1 in the manuscript is labeled as VAMP2 in the data and figure, CYFIP2 is CYFP2 in data, MAPK3 is MK03 in data table and possibly MKO3 in figure, EEF1G is EF1G in supplement, CHD13 is CDH13 and CAD13 in data, DNM2 to DYN2, CTSB to CATB, EB3 to MARE3, CPE is difficult to identify in supplement. For proteins specifically highlighted in the manuscript it would be helpful to provide the full names.

Authors' Response: We thank the reviewer for this comment. Many proteins have multiple names that are used to reference them in the literature, therefore, we decided to be consistent with nomenclature throughout the study and now specifically use the UniProt Identifier for identification throughout the manuscript, figures, and supplementary tables.

Second, the reviewer brings up an important point about the conversion between protein names and gene names, which was a source of the lack of clarity in the original submission. We have now addressed this more clearly within the methods and results sections, specifically noting where these protein to gene name conversions were done for analyses. We have also added a sentence to denote this conversion in each figure legend where this approach was used.

Finally, we have also added a supplementary table that notes the UniProt ID, the full protein name, the associated gene name, and the accession number (**see Supplementary Table 1**) so that it is easy for readers to cross reference. This was critical to improving clarity and transparency and we thank the reviewer for bringing this to our attention.

5. The MS/MS raw data/proteomic data should be uploaded to a freely available database such as ProteomeXchange.

Authors' Response: While the manuscript was under review, we submitted the raw proteomic data to PRIDE. The manuscript now notes "Data are available via ProteomeXchange with identifier PXD022832". The data will be released and made freely available to the public immediately upon acceptance.

6. Full proteomics data should be provided as supplement such as ranked lists and p-values used for RRHO analysis, full GO data including member proteins for each GO term and proteins significantly regulated in analyses. This would especially help with interpretation of Fig 4F, were all significant GO terms included in Fig 4D and 4E which are summarized in Fig4F?

Authors' Response: This comment brings up several important points that have prompted us to revise the figures and add additional supplementary tables. First, we have added multiple tables that include the lists that were used to determine the gene ontology (GO) terms (**Supplementary Tables 2-4**) – and the output lists from the gene ontology analysis as supplementary tables (**Supplementary Tables 7-14**). Because we used Revigo to collapse GO terms with significant overlap we have uploaded both the original list from gProfiler showing all of the GO terms and the associated proteins (**Supplementary Tables 7-8, 11-12**), as well as the Revigo list that was ultimately used to make the final GO figures (**Supplementary Tables 9-10,13-14**). We have done this for both the baseline comparisons in Figure 2 and for the cocaine comparisons in Figure 4. We also have uploaded the ranked lists that were used to generate the RRHO plots in Figure 5 (**Supplementary Tables 15-18**).

Next, we removed figure 4F from the manuscript. Because of the nature of GO analysis, we did not want to make large sweeping conclusions about whether proteins had similar functions, given the broad identifiers of some of the overlapping GO terms. We felt that studying these further in subsequent manuscripts would allow for more powerful and accurate comparisons of these proteins.

7. Additionally, was the RRHO analysis conducted with the p-values? Please indicate in methods. A little more explanation of the differences in interpretation between the four quadrants for the RRHO analysis would aid in the interpretation of Fig5. Authors conclude that larger single peak p-values indicate a larger effect size, is this typically used with the RRHO method?

Authors' Response: This comment was critical to us better clarifying the meaning of the findings with the RRHO plots. To address this, we have revised the results section to further explain how comparisons are made and elaborate on how we interpret these findings. The RRHO analysis was conducted using the Log2FC relative to the sex-specific saline control, which means that the larger p-values presented in the RRHO output do denote more significant overlap determined by relative effect sizes in the differential expression of proteins at those points.

8. More details for the network analysis are important for clarity. What is the rationale for the STRING network analysis? Was the network in Fig 2F only created with significantly regulated proteins? It seems as though 2G and 2H are proteins only from the protein binding GO term and neuron projection term overlaid on the same network in 2F. This is not clear in the manuscript. Were these GO terms significantly overrepresented in the dataset? The manuscript mentions histone proteins and VAMP2/CYFP2/ACTB but not protein binding or neuron projection GO terms for these figures, are these hubs statistically significant or interesting highlights from the authors? Figures 2F, 2G, and 2H are difficult to read the individual protein names, particularly with more negative log fold changes (dark blue symbols).

Authors' Response: To address this comment we have: (1) Added a supplementary table including the

protein names from the STRING analysis and noted this in the figure legend (**Supplementary Table 6**) (2) Made the figure text easier to read, and (3) Expanded the results section to clearly highlight our goals with presenting these data. However, we would also like to note that we are open to removing these analyses from the manuscript if the reviewer and editor think that they are too descriptive and detract from the overall message of the story.

To clarify our goals with these figures, we included them as an additional descriptive element to show that many of the proteins that are significantly different between males and females at baseline interact with one another. For this figure we used STRING analysis to determine the interactions between these significantly differentially expressed proteins, and then for that same network highlighted some of the GO terms that made up a bulk of these proteins. Interestingly most of these proteins could be separated into “Protein Binding” and “Neuron Projection” which is why we chose to highlight the proteins that made up those terms overlaid on the original network.

Further, in the manuscript we noted some of the synaptic proteins involved in release and histone proteins involved in transcriptional regulation as they were interesting to us based on our previous work showing sex differences in neurotransmitter release regulation and cocaine-mediated transcriptional regulation in the same brain region. Therefore, we did not intend to discuss all of the proteins, but rather some that stood out to us as potentially important to us based on past and ongoing work. We specifically note this now in the revised text.

9. Authors should clarify at the beginning of the results that mice are used throughout the study except for the behavioral economics analysis in Fig 1 and include a brief rationale for the use of rats in the results section prior to those data.

Authors’ Response: We have revised the results section to more clearly describe when rats vs mice were used in behavioral studies. Briefly, we used rats for the behavioral economics studies as these tasks are complex and rats are better suited to complete these within-session economic tasks which allow us to gain a more nuanced understanding of what behavioral processes are specifically different between the sexes.

10. These authors have previously published with behavioral economics, however additional methodological details are necessary to interpret this portion of the study. How are alpha and Q0 calculated for each animal? The methods section under “within session threshold procedure” should indicate rats in the title and also states Q0 is a measure of cocaine consumption rather than sucrose.

Authors’ Response: We want to thank the reviewer for noting this, this was a typo as Q0 is a measure of sucrose consumption in this case. We have revised the results and methods sections as suggested to describe behavioral economics analysis more clearly and proofread to make sure that the methods and results refer to sucrose and not cocaine.

We have also expanded the methods section to explain how we fit the curves for each animal, what parameters we are using, and how they are calculated.

11. For the results of the rat behavioral economics experiment, why are Pmax for representative male and female rats in Fig1F and 1G beyond the possible price available to the rats and beyond the x axis?

Authors’ Response: Thank you for noting this point, which prompted us to more clearly explain how these values are calculated and to edit the figures to clarify this point. First, the x-axis for the representative plots was plotted as response/pellet. The P_{max} is the standardized P_{max} value, which is determined by the equation $P_{max} * Q0$. This is important as it allows for a comparable price that is not influenced by the level of consumption across animals and is standardized so that it is comparable across groups and species. The lack of clarity was because our Y axis on the group data did not denote how the standardized P_{max} was calculated. It is now noted that we are plotting the standardized P_{max} on the figures and in the legend. Further, as noted in response to comment 10 we have expanded our description of how each measure was calculated in the methods section.

12. If 11 bins were used, why are 8 data points shown in these figures?

Authors' Response: Each animal was run through 11 bins in the behavioral task; however, if there are no reinforcer deliveries in some of the bins - which often happens in the higher effort bins - they are not used to fit the demand curves. Therefore, in the representatives there are fewer points plotted. Demand curves are generated by curve-fitting individual animals' intake using the equation: $\log(Q) = \log(Q_0) + k \times (e^{-\alpha \times Q_0 \times C} - 1)$ (Christensen *et al*, 2008; Hursh and Silberberg, 2008; Oleson *et al.*, 2009). P_{\max} is determined as the first derivative point where the slope of the function = -1. Therefore, if an animal does not respond in the bins following these points (i.e. the later, higher effort bins) it does not influence the calculation. This is expanded upon and clarified within the methods.

13. Is Q0 statistically different between male and female rats?

Authors' Response: This is an important comment that prompted us to add additional data to the manuscript. We analyzed the data for consumption and now report the values in the methods and as a supplementary figure (**Rebuttal Figure 1**). There was no significant difference between males and females in this measure reported as total consumption or when normalized for body weight.

14. For clarity it would help to add a line for Q0 and a line for Pmax in the representative curves.

Authors' Response: We have added a line for Q0 and Pmax on each of the curves, which we agree improves clarity.

15. Methods section states that animals are food restricted during self-administration- is this sucrose or cocaine SA? The heading "operant conditioning" only includes information for mice, does this information also apply to the rats?

Authors' Response: We have updated the Methods section to address this lack of clarity. All animals were food restricted in these studies (for both cocaine and sucrose self-administration), except for the experiments where we were testing sated self-administration of sucrose in mice (in **Figure 1B**). We have clarified this in the figure legend as well.

16. The introduction states that there is "almost non-overlapping molecular plasticity induced by cocaine self-administration between males and females" and the discussion states that "cocaine self-administration induces non-overlapping protein expression patterns in males and females", however the results show that there is minimal overlap in individual statistically significant proteins, but concordant effects when examining the proteomics overall without arbitrary cutoffs. This is a strength of this manuscript and the introduction and discussion should reflect these results. The discussion also states that the RRHO analysis shows that cocaine alters the proteome in the NAc but does so differently in males and females, this seems opposite from the results.

Rebuttal Figure 1. (*manuscript Supplementary Figure 1*)

(left) Sucrose consumption in total milligrams during the last session of training on a fixed-ratio 1 schedule of reinforcements.

(right) Sucrose consumption plotted as milligram per kilogram body weight in males and females.

*one female was a statistically significant outlier and was removed from analysis; this did not change the statistics and both measures were not significant either way.

Authors' Response: We agree that we could have done a better job of capturing the complexities of these results in the abstract, introduction, and discussion and have edited these to better explain these data in a clear and accurate way.

17. The results state that “using odds ratio analyses to determine the probability of overlap and whether it differs from what would be predicted if there were no effect” appears to be a fragmented sentence. Also, “it is possible that males and females have similar cocaine-associated behavioral profiles being mediated different molecular mechanisms” also seems to be fragmented.

Authors' Response: We have made these corrections and have thoroughly reviewed the manuscript to identify other grammatical/formatting errors.

18. Do the odds ratio analyses use all detected proteins or only significantly altered proteins? Full statistics should be included for all of the odds ratio analyses.

Authors' Response: The odds ratio comparisons use the entire proteomic data list. However, we distinguish proteins from “significantly altered” and “not significantly altered”. We have clarified these distinctions in the results and methods. Moreover, we have included additional statistical results of the odds ratio testing.

19. Are there other interpretations of the final experiment where cocaine administration is blocking the sex differences in sucrose consumption? Were there any differences in locomotor activity after cocaine i.p. compared to the previous sucrose consumption experiment?

Authors' Response: This is an interesting point that we had not considered. While we did not measure locomotion following I.P. administration, it is important to highlight that repeated I.P. cocaine injections were administered in the home cage as opposed to a novel context that is typically used for cocaine sensitization studies. In fact, previous work has shown that sensitization is context-dependent and only occurs when cocaine is administered in a novel environment (Post et al., 1981; Vezina et al., 1989; Post et al., 1992); thus, we do not hypothesize differences in our model. However, we did not explicitly test this and would be interested to know how motor sensitization manifests between males and females.

20. The discussion states that of the five overlapping proteins regulated by cocaine, only two are regulated in the same direction, but results show 3.

Authors' Response: Reviewer 2 is correct, and we have corrected this typo in the updated manuscript.

21. In the description for Fig3, panel D states that female mice show higher rate of behavior at higher cocaine dose than males but results state this is a trend.

Authors' Response: Thank you for highlighting this inconsistency, we have corrected the manuscript to correctly state the statistical trend.

22. Fig4 panel B legend has total twice in a row. Methods section motivational testing paragraph has “until stable and then and subsequently”. DTT should be dithiothreitol and IAM should be iodoacetamide.

Authors' Response: Thank you for bringing these errors to our attention - we have corrected them in the revised version of the manuscript.

Reviewer #3:

1. This study examined protein expression patterns using large-scale proteomic analysis of the nucleus accumbens (NAc) from one set of naïve male and female mice and one set of male and female mice 24-h after a 10-d cocaine self-administration regimen. Behaviorally, there were no robust sex differences in cocaine self-administration behavior. However, in a sucrose self-administration paradigm, cocaine-naïve female mice (and rats) showed greater motivation-like behavior for sucrose compared to cocaine-naïve male mice (and rats). The proteomic data showed significant sex differences in the regulation of specific proteins and protein pathways in both naïve and cocaine-exposed mice – although GO analysis showed that cocaine regulates proteins with similar functions in male and female NAc. Not only were different proteins regulated in a sex-dependent manner, but the directionality of many proteins was different (e.g., upregulated in males and downregulated in females). The crux of the paper is the finding that cocaine-induced changes in protein regulation was such that it normalized male vs female protein levels under naïve conditions. In other words, if a protein was significantly reduced in male NAc compared to female NAc in naïve mice, that protein was likely to be upregulated in cocaine-exposed males (thus normalizing that protein to female levels). A partial test of this finding was done in which separate male and female mice were treated non-contingently with cocaine and then tested for motivation to self-administer sucrose. Whereas naïve female mice show greater motivation to self-administer sucrose, this sex difference was eliminated after cocaine injections. Although the paradigms are not exact, these data are consistent with the idea that cocaine-induced normalization of protein expression in the NAc also equalizes male and female sucrose self-administration.

Authors' Response: We would like to thank the reviewer for their comprehensive description of the study. The critical comments provided by the reviewer were instrumental in improving the clarity and transparency of the study and we are incredibly appreciative.

2. This paper has many strengths, and there is a great deal of important, useful, novel, and potentially significant data. The combination of techniques and the use of powerful bioinformatic methods to address the question of how cocaine alters NAc plasticity and motivated behaviors are strengths. However, some of the strengths are weakened by assumptions of what the data represent, which is partly due to a lack of all necessary controls and partly due to inaccurate use of language to describe the results. This major issue and several other comments are described below.

Authors' Response: We would like to thank the reviewer for their positive comments about the impact of this study, as well as the suggestions for clarification and improvement which we have used to revise the manuscript.

3. In the Introduction (and elsewhere), clinical and preclinical findings specific to psychostimulants are generalized to all drugs of abuse. For example, within lines 79-83, it is stated "...when drug use is taken into consideration, female use exceeds that of males..." This is fairly specific to psychostimulants, and that should be made clear. A common and unfortunate consequence of generalizing is that incorrect assumptions are made by readers. Also, in the Discussion (lines 330 – 332), it is stated "...increased sensitivity to drugs of abuse seen in females across species." Please reword these types of generalizations to be more accurate.

Authors' Response: We agree with the reviewer that it is critical to be more specific when referencing previous findings on sex-specific drug effects. In our revised manuscript we have clarified that we are commenting on the effects of psychostimulants, specifically.

4. Sucrose is used as a reinforcer and is described as a "natural reward". I disagree that it is a natural reward, and this has been discussed in much of S. Ahmed's work (see Lenoir et al., PLoS One, 2007). It doesn't change the value of testing sucrose reward in the current study, but simply refer to it as a

“reward” or a “reinforcer” or something similar.

Authors’ Response: This is an important point. We have changed all references of sucrose as a natural reward to be described as either a “reward” or a “reinforcer” depending on the schedule/context under which it was presented.

5. Given #2, it isn’t accurate to refer to behavior in response to sucrose reward as “baseline”. Rather, there is sucrose self-administration behavior and cocaine self-administration behavior. These can be compared – but not as cocaine vs baseline reinforcement.

Authors’ Response: Thank you for noting this. We agree with the reviewer and have changed the language regarding the sucrose-reinforced behavior which more accurately represents the data.

6. Throughout the manuscript, there are examples in which the language overstates the current findings or findings from other groups. For example, in lines 120-121, the statement “...characterize the molecular mechanisms regulating sex-specific reward behaviors in male and female mice” overstates what this paper does. There is no direct evidence presented that a particular molecular mechanism regulates sex-specific reward behaviors. There is indirect evidence.

Authors’ Response: We agree, and we have softened the language here and throughout the manuscript.

7. I do not see the number of animals used for each study/treatment group written anywhere. Please add.

Authors’ Response: We have amended the Methods section to specifically state the sample size in each experiment. Further, we have included individual data points on all of the graphs within the main and supplementary figures.

8. In Figure 1, graph D and its interpretation are confusing. If D is the change in sucrose from VR3 to VR5, and males and females had similar active responses in graph C, then it isn’t clear how there could be such a difference in graph D. The text (line 137) that males trended towards decreased consumption from VR3 to VR5. Again, I don’t see how that is possible given graph C.

Authors’ Response: Thank you for bringing up this important point. This is for two reasons. First, the variable ratio schedule means that the relationship between reinforcers delivered and responding is not always exactly linear. Second, Figure 1D is represented in an increase in milligrams consumed per gram of body weight. Because females are significantly smaller than males, a similar increase in consumption between the sexes would result in a larger change in this measure in females. We think that this is particularly important as it controls for the body weight differences that could influence total consumption and would be independent of the motivational measures that were probed in these studies. We now specifically noted this in the figure legend and results.

9. Related to #6, the conclusion is made that females are “less sensitive to changes in cost than males”. But if the males are not really changing their consumption from VR3 to VR5 whereas females are upping their consumption, isn’t it more accurate to say that females are MORE sensitive to changes in cost, and adjust their behavior accordingly? Alternatively, males and females could be similarly sensitive, but males respond by reducing effort and females respond by increasing effort. I might not understand the concept of sensitivity in this context – some explanation is warranted.

Authors’ Response: Again, we thank the reviewer for noting this. When we describe females as insensitive to cost, we intended to refer to the phenomenon that with increased costs, females will continue to consume the

same amount (or more) of sucrose by increasing their effort. In other words, changes in cost do not appear to reduce consumption in females. However, we realize that this could be confusing as the increase in schedule coincided with increased consumption, which could be noted as sensitive to cost in the opposite way than we intended.

To address this, we have revised our wording of this in the manuscript and now use the economic experiments that we conducted subsequently to talk about cost sensitivity in behavioral economics terms - where they are clearly operationally defined in the field- so that the findings are clear and do not lead to confusion.

10. A major weakness, alluded to in the beginning of this review, is that the different cohorts of mice are treated in different ways, and yet direct comparisons are made between the groups. Specifically, proteomic analysis was done on NAc tissue from naïve male and female mice (referred to as “baseline”) and on male and female mice that underwent cocaine self-administration and 24-h withdrawal. Results from those two cohorts were compared, and differences attributed to cocaine self-administration. But there are several other differences in the groups: naïve vs experience/self-admin behavior in the operant chambers, no surgery vs catheter surgery, minimal handling vs daily handling. Given these differences in the cohorts, there is rigor and validity to the reported sex differences in proteomics of (1) naïve mice and (2) mice that self-administered cocaine and had 24-h of withdrawal. However, it is not rigorous enough to conclude that the changes in protein expression after cocaine self-administration “eliminate” those differences that exist in naïve mice. Given that this is the primary conclusion of the paper, either an appropriate control group needs to be run (ideally mice that self-administer saline under the same conditions as the cocaine group) and the results compared to the cocaine self-admin group, or the conclusions of the paper need to be changed to reflect this issue.

Authors’ Response: We apologize for the lack of clarity. The control mice – which were used for the original baseline comparisons in **Figure 2** - did undergo saline self-administration and are thus the appropriate control as the reviewer suggested.

This is an important consideration and we have now added an additional section in the methods to explain the behavioral groups for the proteomics design more clearly and explicitly. First, all of the animals were the same age, had identical experience in our facilities, samples were prepared in tandem, and mass spec was done all at the same time in one large experiment. This allowed for us to reduce the variability from other factors.

Second, the ‘baseline’ comparisons are done between our male and female saline groups. These animals were catheterized and placed into the operant chambers each day but did not get any cocaine. We have shown previously that there are not significant differences between saline controls and naive controls, which allows us to use the saline group to both define the baseline sex differences and accurately compare the effects of cocaine on these differences within cohort and group. We think that this design is a particular strength of the current study as it allows for these complex computational comparisons, that could not be done with other designs.

11. Similarly, in the proteomic study from mice that underwent cocaine self-administration and 24-h withdrawal, it is not accurate to assume the protein changes are due to cocaine self-administration. Changes in protein expression and plasticity could also arise from acute cocaine effects from the last cocaine self-admin session or from withdrawal effects. In this case, changing the text throughout the manuscript will suffice. It should not be stated that any proteomic changes are due to cocaine self-administration. It is ok to say “prior cocaine exposure” or “cocaine self-administration and 1 day withdrawal”, etc.

Authors’ Response: We thank the reviewer for bringing up this point, which we have now expanded upon in the manuscript. We have extensive experience with studies focused on the pharmacokinetics of cocaine and

how this affects neural plasticity within reward circuits. These studies show that within 60 minutes following the last i.v. injection, cocaine levels in the brain are not detectable (**Rebuttal Figure 2**; Calipari et al., 2014, Zimmer et al., 2012). Thus, the observed effects are not likely to be due to the acute effects of cocaine.

Nevertheless, the point that the reviewer makes is well taken. We intended to make the point that the changes occur because of the previous experience of the animal (cocaine self-administration and acute withdrawal). Therefore, we have softened our wording in this regard and now state “prior cocaine exposure” to be clear about what we are studying.

12. The behavioral experiment shown in Figure 5 is based on the idea that cocaine self-administration “eliminates” baseline sex differences in proteomics in the NAc, and this in turn eliminates sex differences in reward-seeking. This is supported by the finding that prior cocaine treatment results in males and females showing similar sucrose-seeking behavior (as opposed to Figure 1). Again, there is an issue with consistency of methods. The entire paper focuses on the effects of cocaine self-administration on protein expression, and yet the experiment in Figure 5 uses non-contingent cocaine injections.

Authors’ Response: We agree with the reviewer that the difference in route of cocaine administration is an important factor. However, as training mice to operantly respond for cocaine will confound subsequent operant responding for sucrose if it is done in the same context, we decided experimenter-administered cocaine would provide a more comparable experiment to the original study presented in Figure 1. However, to address the Reviewer’s concern we have toned down the language to properly frame the conclusions from the study in Figure 5.

We are also designing experiments that would allow us to assess these effects and potential neural mechanisms underlying them; however, we feel that this is outside of the scope of the current study.

13. Taking points 8-10 together, the line of reasoning and conclusions of the paper are flawed as written. They might be totally accurate, which would be fantastic. But the experiments were not conducted in a way that allows for those conclusions to be made.

Authors’ Response: We believe that the clarifications and modifications we have made (see comments on issues 8-10) address the core concerns and do not fundamentally alter the core conclusions of the manuscript. We have revised our wording to be more accurate and have clarified that the groups that were used to make comparisons in the manuscript.

14. Lines 229– 232: The statement “...we found there is a significant overlap between the proteins unchanged by cocaine in males and those unchanged by cocaine in females (compared to predicted overlap by random chance), but no significant overlap in proteins changed by cocaine in males and those changed by cocaine in females (Fig. 4C)” isn’t strongly supported by the odds ratio data. Even

Rebuttal Figure 2. (not included within the manuscript)
*Adapted from Calipari et al., 2014.

Equations from Pan et al., 1991 were used to model brain levels of cocaine from representative animals within a self-administration session over time. Data is expressed as arbitrary units.

Modeled brain levels of cocaine (y-axis) versus time (x-axis) throughout the entire session for an individual rat self-administering cocaine to highlight the fluctuation of cocaine levels within the brain.

ShA denotes a “short access” session, which is 2 hours in length. This results in high, sustained brain cocaine over the 2-hour session that rapidly decline following session termination.

though there was no significant overlap in proteins changed by cocaine in males and those changed by cocaine in females, the odds ratios for that comparison and the comparison of proteins unchanged by cocaine in males and those unchanged by cocaine in females are almost the same (~2.4 vs ~2.8). Some mention and discussion of this is warranted.

Authors' Response: We thank the reviewer for bringing attention to this issue. While the values of the odds ratio comparisons are comparable, the odds ratio comparison does not reach significance in the proteins significantly changed by cocaine. We now mention this in the revised manuscript.

15. There needs to be validation of at least some of the protein changes determined with the mass spec. It would suffice to choose one or two proteins that are robustly up or down regulated in each sex/group and validate using western blots or some other method. As written, the entire proteomic data sets appear to come from one cohort of naïve mice and one cohort of mice that self-administered cocaine. This is not sufficient and does not instill confidence that the results would be replicable.

Authors' Response: With the advent of large-scale -omic analysis, it is critical to evaluate the reliability and replicability of the data collected. Several studies have demonstrated that mass spectrometry provides greater specificity and reliability versus other protein detection methods, such as Western blotting, as mass spectrometry directly detects peptides using mass and charge whereas Western blotting indirectly determines quantitative levels of protein using antibodies which are subject to error in the form of non-specific or incomplete binding, or any number of technical errors introduced in the blotting process. However, we also agree that -omic data should be supported with more targeted approaches. We have included supplementary data validating some detected changes using RT-qPCR to show that there is regulation of these targets at both the mRNA and protein level (**Rebuttal Figure 3**) - thus adding additional validity

Rebuttal Figure 3. (manuscript Supplementary Figure 3)

(D) DDX3X (gene *Ddx3x*) was increased in females versus males at the (i) protein and (ii) transcript level.

(E) DDX3Y (gene *Ddx3y*) was increased in males versus females at the (i) protein and (ii) transcript level.

(F) CYFP2 (gene *Cyfp2*) was increased in females versus males at the (i) protein and (ii) transcript level.

(G) Expression of *Ddx3x* is higher in the NAc of female mice compared to male but was not changed by cocaine experience.

(H) Expression of *Ddx3y* is higher in the NAc of male mice compared to female mice but was not changed by cocaine experience.

(I) Expression of *Cyfp2* is increased in the NAc of female mice compared to male mice but was not changed by cocaine experience.

Data reported as mean ± S.E.M.* p<0.05, ** p<0.01, *** p<0.001, ****p<0.0001. Sal = saline group, coc = cocaine group

to our findings as well as a better understanding of the mechanistic basis for these changes.

16. Please clarify if the cocaine dose-response experiment was initiated after the mice were done with the FR5 sessions at cocaine (1 mg/kg/infusion). It is advised that there be an extinction period after the 1 mg/kg self-admin period followed by demonstration that the mice can differentiate between vehicle and cocaine. This gives confidence in the dose response function. As it is displayed in Figure 3C, how do we know that the high number of active responses for the low 0.1 mg/kg dose is not just the mice seeking drug because they are used to getting 1.0 mg/kg and now they are suddenly getting 1/10th that dose?

Authors' Response: We agree with the reviewer that specificity of operant tasks is critical in extrapolating conclusions. In our paradigm the doses are scrambled for each animal, precluding the effects of training with 1mg/kg on responding specific to 0.1 mg/kg; thus, we do not think that this contributes to the observed effects. Further, an important characteristic of cocaine self-administration across species is a lack of a clear ascending limb of the dose response curve under most conditions. Jim Woods' lab has spent a considerable amount of effort explicitly outlining the precise conditions under which this ascending limb occurs - which is only under certain conditions that are not present in most traditional dose-response paradigms (Flory and Woods, 2003). Because of this, we agree with the reviewer that presenting the data in this fashion can make it difficult to interpret. To address this, we have converted the data into demand curves (consumption of cocaine as a function of the price in responses/mg; **Rebuttal Figure 4**). We then fit these demand curves and conducted a behavioral economic analysis to define Q_0 and P_{max} between males and females.

The reason that this analysis avoids the issues that the reviewer brought up is that P_{max} is determined as the first derivative point where the slope of the function = -1 (Christensen *et al*, 2008; Hursh and Silberberg, 2008; Oleson *et al.*, 2009). Therefore, the later, higher effort bins (or the low doses in question in this case) do not have a large influence on the calculation of consumption (Q_0) and motivation (P_{max}) in isolation. Thus, this gives a more comprehensive measure of the effect without a large influence from variable individual data points or behavior that is controlled by lower doses in isolation.

When we did this analysis, we found – as the reviewer hypothesized - that there are not robust sex differences in cocaine consumption or motivation (**Rebuttal Figure 5**). We now present these data in the manuscript in place of the other response curve and report it as such. We have left the other analyses in the paper as a supplementary figure (**Supplementary Figure 3**), but we believe plotting the data in these ways will allow the reader to get a full picture of the data and where sex differences do and do not appear to emerge. In fact, these data are consistent with our previous work in rats showing that there are not robust sex differences in P_{max} and Q_0 for cocaine under most conditions (Johnson *et al.*, 2019).

Rebuttal Figure 4. (not included within the manuscript)

A concentration-response curve was run across days with doses counterbalanced between animals (0.1, 0.3, 1, 3, mg/kg). Data was plotted as a demand curve where consumption was plotted on the y axis and price in responses to obtain 1 mg cocaine was plotted on the x axis. Curves were fit to determine consumption at a minimally constraining price (Q_0) and maximal price paid (P_{max}) in males and females. Standardized P_{max} ($Q_0 * P_{max}$) was calculated to allow for comparisons that are not influenced by the relative level of consumption and are comparable across groups.

Rebuttal Figure 5. (manuscript Figure 3)

(A) A series of behavioral experiments were run to assess sex differences in motivation for cocaine self-administration in males and females. Schematic/timeline of self-administration.

(B) Average responses for cocaine under escalating fixed ratio schedules in male and female mice. Male and female mice acquire and consume cocaine at comparable rates under FR1, 3, and 5 schedules of reinforcement.

(C) A concentration-response curve was run across days with doses counterbalanced between animals (0.1, 0.3, 1, 3, mg/kg). Data was plotted as a demand curve where consumption was plotted on the y axis and price in responses to obtain 1 mg cocaine was plotted on the x axis. Curves were fit to determine consumption at a minimally constraining price (Q0) and maximal price paid (P_{max}) in males and females. Standardized P_{max} ($Q0 \times P_{max}$) was calculated to allow for comparisons that are not influenced by the relative level of consumption and are comparable across groups.

(D) Q0 – plotted as mg/kg to control for body weight differences – was not significantly different between males and females.

(E) Standardized P_{max} was not significantly different between males and females.

We want to thank the reviewer for this comment – as well as the other comments throughout this review - as we believe that the new analysis is much more rigorous, avoids several confounds, and provides a clear and operationally defined account of the data.

References

- Calipari ES, Ferris MJ, Siciliano CA, Zimmer BA, Jones SR. Intermittent cocaine self-administration produces sensitization of stimulant effects at the dopamine transporter. *JPET*. 2014. 349(2):192-8
- Christensen C, Silberberg A, Hursh S, Huntsberry M, Riley A. Essential value of cocaine and food in rats: tests of the exponential model of demand. *Psychopharmacology*. 2008; 198:221–229.
- Flory and Woods. The ascending limb of the cocaine dose-response curve for reinforcing effect in rhesus monkeys. *Psychopharmacology*. 2003. 166(1):91-4
- Hursh SR, Silberberg A. Economic demand and essential value. *Psychol Rev*. 2008. 115(1): 186-98
- Johnson AR, Thibeault KC, Lopez AJ, Peck EG, Sands LP, Sanders CM, Kutlu MG, Calipari ES. Cues play a critical role in estrous cycle-dependent enhancement of cocaine reinforcement. *Neuropsychopharmacology*. 2019; 44(7):1189-1197.
- Oleson EB, Richardson JM, Roberts DCS. A novel IV cocaine self-administration procedure in rats: differential effects of dopamine, serotonin, and GABA drug pre-treatments on cocaine consumption and maximal price paid. *Psychopharmacology*. 2009. 214(2):567-77
- Pan HT, Menacherry S, Justice JB. Differences in the pharmacokinetics of cocaine in naive and cocaine-experienced rats. *J Neurochem*. 1991; 56:1299–1306
- Post RM, Lockfeld A, Squillace KM, Contel NR. Drug-environment interaction: context dependency of cocaine-induced behavioral sensitization. *Life Sci*. 1981; 28:755–760.
- Post RM, Weiss SR, Fontana D, Pert A. Conditioned sensitization to the psychomotor stimulant cocaine. *Ann N Y Acad Sci*. 1992; 654:386–399.
- Vezina P, Giovino AA, Wise RA, Stewart J. Environment-specific cross-sensitization between the locomotor activating effects of morphine and amphetamine. *Pharmacol Biochem Behav*. 1989;32:581–584.
- Zimmer BA, Oleson EB, Roberts DCS. The motivation to self-administer is increased after a history of spiking brain levels of cocaine. *Neuropsychopharmacology*. 2012. 37(8):1901-10

REVIEWERS' COMMENTS:

Reviewer #1 (Remarks to the Author):

The edited manuscript is fine as submitted. The authors did everything asked by the reviewers. The manuscript is greatly improved.

Reviewer #2 (Remarks to the Author):

The authors have clearly addressed all of my concerns and have edited the manuscript accordingly including adding supplementary tables, modifying analyses, and adding new experiments as well as submitting the proteomics data to PRIDE.

Reviewer #3 (Remarks to the Author):

The revised manuscript and rebuttal letter are thorough, thoughtful, and addressed all concerns raised in my original review. The responsiveness of the revision to the critiques and the decision to analyze/present data in different formats - sometimes resulting in different interpretations - is commendable for transparency. All of my concerns were addressed quite satisfactorily.